# Pre-training under infinite compute

**Konwoo Kim$^\infty$, Suhas Kotha$^\infty$, Percy Liang, Tatsunori Hashimoto**
Stanford University

## Abstract

Since compute grows much faster than web text available for language model pre-training, we ask how one should approach pre-training under fixed data and no compute constraints. We first show that existing data-constrained approaches of increasing epoch count and parameter count overfit, and we improve upon such recipes by tuning regularization, finding that the optimal weight decay is $30\times$ larger than standard practice. Since our regularized recipe monotonically decreases loss following a power law in parameter count, we estimate its best possible performance via the **asymptote** of its scaling law rather than the performance at a fixed compute budget. We then identify that ensembling independently trained models achieves a significantly lower loss asymptote than the regularized recipe. Our best intervention combining epoching, regularization, parameter scaling, and ensemble scaling achieves an asymptote at 200M tokens using $5.17\times$ less data than our baseline, and our data scaling laws predict that this improvement persists at higher token budgets. We find that our data efficiency gains can be realized at smaller parameter counts as we can distill an ensemble into a student model that is $8\times$ smaller and retains $83\%$ of the ensembling benefit. Finally, our interventions designed for validation loss generalize to downstream benchmarks, achieving a $9\%$ improvement for pre-training evals. Our results show that simple algorithmic improvements can enable significantly more data-efficient pre-training in a compute-rich future.

## 1 Introduction

Language model pre-training has historically been studied under compute constraints at training (Kaplan et al., 2020; Hoffmann et al., 2022) and inference (Brown et al., 2024; Snell et al., 2024) while assuming access to unlimited web text. However, web data grows by $1.03\times$ per year, whereas compute spent on pre-training grows by $4\times$ per year (Villalobos et al., 2024; Sevilla and Roldán, 2024). In anticipation of a regime where compute vastly exceeds data, we ask:

*How should one approach pre-training under fixed data and no compute constraints?*

To establish a baseline, we fix a seed training corpus of 200M tokens of web text and evaluate a *standard recipe* following existing data-constrained approaches of repeating data (Muennighoff et al., 2023) and increasing parameter count (Kaplan et al., 2020) (Section 2). We find that either too many epochs or too many parameters results in the loss eventually increasing due to overfitting. This bounds the performance improvements we can get from tuning this recipe, even if we were willing to spend more compute in exchange for a better model.

We instead get predictable monotone scaling in parameter count by considering a *regularized recipe* (Section 3). Currently, regularization used in pre-training is often adopted from existing recipes, defaulting to a weight decay of $0.1$ from Brown et al. (2020). We find this amount to be inadequate for preventing overfitting under data constraints as the optimal weight decay is $30\times$ larger than standard practice for our most over-parameterized models. After jointly tuning weight decay, learning rate, and epoch count at each parameter count $N$, loss closely follows a power law in $N$ for parameter-to-token ratios $140\times$ larger than Chinchilla, as shown in Figure 1.

Normally, we would compare two recipes by evaluating performance at different train or inference compute budgets (Hoffmann et al., 2022; Snell et al., 2024). However, this does not reflect our

---

$^\infty$Equal contribution.

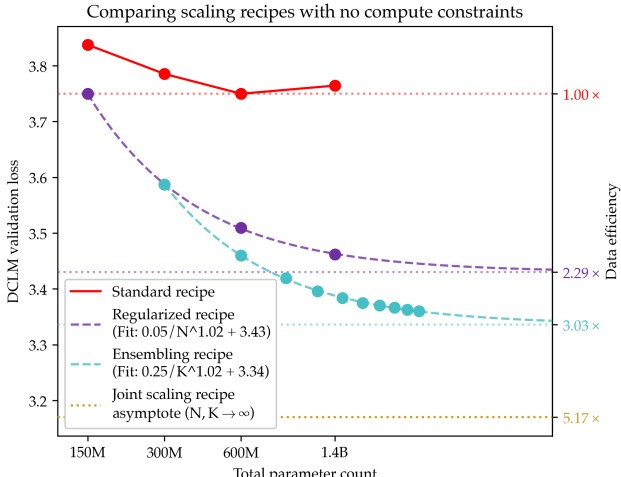

Figure 1: **Comparing scaling recipes with no compute constraints.** To simulate a data-constrained future, we restrict models to 200M tokens. (1) Standard recipes overfit with too many epochs or parameters, even if we tune the epoch count at each parameter count $N$ (2) By correctly tuning regularization for each $N$, loss monotonically decreases following a power law in $N$. We predict the best possible loss of the regularized recipe by the **asymptote** of its power law. (3) Instead of scaling $N$, we achieve a lower asymptote by ensembling $K$ models of size 300M as $K \to \infty$. (4) Composing parameter and ensemble scaling improves the asymptote, and we estimate that the baseline would need $5.17\times$ more data to match its loss, even with infinite compute. These data efficiency wins hold for larger token counts, distilled models, and downstream benchmarks (Sections 5, 6, 7).

interest in the best possible performance under fixed data and no compute constraints. Since the loss of the regularized recipe continues to decrease as $N$ increases, we are interested in the limit of the loss as $N \to \infty$. More generally, we propose evaluating monotone scaling recipes by the **asymptote** of their scaling law (e.g. 3.43 for the regularized recipe as seen in Figure 1). By preferring recipes with lower loss asymptotes, we can train better models at sufficiently high compute budgets.

Though taking the parameter count to infinity is one possible limit under infinite compute, we ask if we can design recipes with even lower asymptotes. We consider an alternative *ensembling recipe* where we average the logits of $K$ independently trained models of the same size (Section 4). The ensembling recipe achieves a lower asymptote as $K \to \infty$ compared to the regularized recipe as $N \to \infty$ (Figure 1). At sufficiently high parameter counts, it is better to train multiple smaller models instead of a single larger model. We further show that ensembling and parameter scaling compose, achieving a lower asymptote when following the *joint scaling recipe* of taking both $K, N \to \infty$.

Since our previous experiments were on the scale of 200M tokens, we study how our recipes scale across higher token counts and find that the asymptotes themselves follow a scaling law (Section 5). Our estimates indicate that the joint scaling recipe achieves its 200M asymptote with $5.17\times$ less data than the standard recipe. Importantly, extrapolation of our data scaling laws indicates that the data efficiency improvements will persist at higher token counts.

Though the asymptotes of our recipes benefit the most from large parameter counts, we find that distillation (Hinton et al., 2015; Kim and Rush, 2016) allows us to retain most of the loss improvements without increasing inference parameter count. Distilling an 8-ensemble into a single 300M model retains $83\%$ of the ensembling loss improvement over the best regularized 300M model and outperforms the asymptote of the regularized recipe. We also find that self-distilling a 300M teacher into a student of the same size reduces loss, improving data efficiency without ever explicitly training a model of higher parameter count.

Finally, we confirm that improvements on validation loss translate to improvements on downstream benchmarks (Section 7). Ensembles with better validation loss perform better on downstream benchmarks, with our best ensemble outperforming our best unregularized model by $9\%$ on average over PIQA, SciQ, and ARC Easy (standard benchmarks for models at our scale (Thrush et al., 2025)).

## 2 STANDARD PRE-TRAINING

Historically, pre-training has focused on training the best possible models subject to compute or parameter constraints. Under train compute constraints, recipes like Chinchilla recommend jointly scaling data and model size with $20\times$ more tokens than parameters (Kaplan et al., 2020; Hoffmann et al., 2022). Under parameter constraints for cheaper inference, current practice opts for over-training language models relative to Chinchilla with token counts $2000\times$ larger than parameter counts or distilling from preexisting larger models (Gadre et al., 2024; Grattafiori et al., 2024; Sardana et al., 2025; Busbridge et al., 2025).

Prior works prescribe such scaling recipes by always training on fresh data. In this paper, we instead study data-constrained pre-training, where we cannot jointly scale data and model size. We analyze the purest form of the problem by lifting all other constraints (including compute) besides data. We formalize standard pre-training as a training routine $\mathcal{A}$ that accepts arguments such as token count $D$, parameter count $N$, epoch count $E$ to produce a model $M$ with loss $\mathcal{L}(M)$. Unspecified arguments are passed through hyperparameter tuple $H$. Our data-constrained pre-training objective becomes $\mathcal{L}_D^* = \min_H \mathcal{L}(\mathcal{A}(D, H))$.

We construct a controlled pre-training environment with a limited amount of web data from DCLM (Li et al., 2025). Since our algorithms spend more compute than Chinchilla scaling at a fixed data budget, we default to 200M tokens and test whether our findings hold across higher token counts in Section 5. For evaluation, we defer to loss on a held-out i.i.d. validation set which is shown to correlate with downstream capabilities in Section 7 and prior work (Chen et al., 2025; Thrush et al., 2025; Gadre et al., 2024). To best represent standard practice, we follow a standard auto-regressive recipe (full details in Appendix B).

### 2.1 EVALUATING EXISTING DATA-CONSTRAINED RECIPES

Since the amount of fresh data is limited, we build a standard recipe of increasing repetition count (Muennighoff et al., 2023) and parameter count (Kaplan et al., 2020). Since there is unlimited compute, we depart from compute-efficient practice by training models that are larger relative to the token count, defaulting to 300M parameter models for 200M tokens.

We first increase the epoch count $E$ at a fixed parameter count, taking $E\times$ more training compute. Figure 2 (left) shows for high $E$, overfitting occurs and loss increases. These findings contradict the functional form of the decay-based scaling law in Muennighoff et al. (2023), which posits that loss monotonically decreases in $E$. Their work acknowledges this discrepancy and removes most overfit runs from their scaling law (see their Appendix D).

Since increasing epoch count arbitrarily hurts loss, we turn to increasing parameter count. To establish a competitive baseline, we jointly tune epoch count and learning rate for each parameter

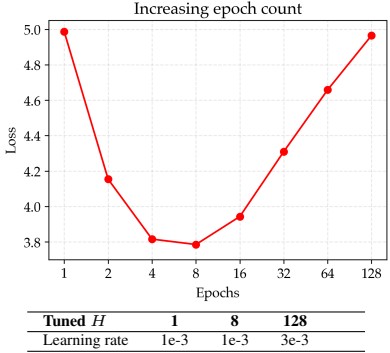

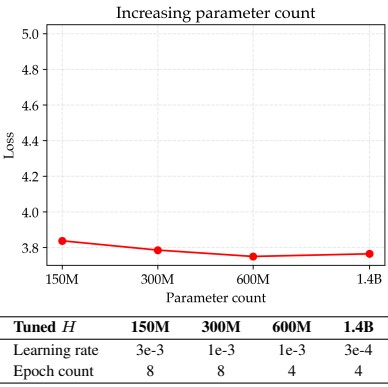

| Tuned $H$ | 1 | 8 | 128 |
|---|---|---|---|
| Learning rate | 1e-3 | 1e-3 | 3e-3 |

| Tuned $H$ | 150M | 300M | 600M | 1.4B |
|---|---|---|---|---|
| Learning rate | 3e-3 | 1e-3 | 1e-3 | 3e-4 |
| Epoch count | 8 | 8 | 4 | 4 |

Figure 2: **Evaluating standard recipe of epoching and parameter scaling for 200M tokens.** Left: Though repeating the data lowers the loss, too many repetitions results in overfitting for 300M models. Right: We try increasing parameter count, tuning the epoch count at each parameter count. We similarly find that loss starts increasing. Moreover, increasing the parameter count $10\times$ improves the loss by less than $0.1$.

count (Appendix C.1). We find minimal improvement in loss with higher model size, with our 1.4B model performing worse than our 600M model. This is consistent with the single-pass findings of Kaplan et al. (2020), Figure 9 which show that increasing parameter count eventually starts increasing loss for fixed data. It is likely that both higher epoch and parameter count result in overfitting the train set, detailed in Appendix C.5.

## 3 REGULARIZED PARAMETER SCALING

We show that to get the best performance from these over-parameterized, epoched models, it is critical to regularize pre-training with much higher weight decay than standard practice. To jointly tune weight decay, learning rate, and epoch count, we perform an extensive search for "locally optimal" hyperparameters using a coordinate descent algorithm inspired by Wen et al. (2025) (details in Appendix C.1). We find that over-parametrized models need much higher weight decay, over $30\times$ larger than the standard practice of $0.1$ (Figure 3, right table).

With this tuning, loss follows monotone scaling in parameter count for models up to $140\times$ larger than Chinchilla as shown in Figure 3. This agrees with theory for over-parameterized regression that predicts that even when loss does not monotonically decrease due to double descent, the loss will monotonically decrease when regularization is optimally tuned (Advani and Ganguli, 2016; Nakkiran et al., 2021; Canatar et al., 2021; Simon et al., 2024). In Appendix C.2, we show how locally-optimal tuning is critical to achieve monotone scaling.

To capture how increasing parameter count improves loss, we fit a power law with an asymptote as $\hat{\mathcal{L}}_{D,N} := \frac{A_D}{N^{\alpha_D}} + E_D$ where we fit free variables $A_D, \alpha_D, E_D$. Our fit across four parameter counts results in $\hat{\mathcal{L}}_{200M,N} = \frac{0.05}{N^{1.02}} + 3.43^1$. The exponent of $1.02$ for parameter scaling is high given that Chinchilla finds a parameter scaling exponent of $0.34$. This suggests that when we better leverage the data, there is faster improvement from larger models.

Our monotone scaling law differs from compute-optimal prescriptions where increasing $N$ can hurt performance due to training on less data. We characterize our best possible performance unconstrained by compute as $\lim_{N \to \infty} \hat{\mathcal{L}}_{D,N}$ e.g. the asymptote $E_D$. The asymptote for the regularized recipe law predicts that the best possible model achieves loss $3.43^2$.

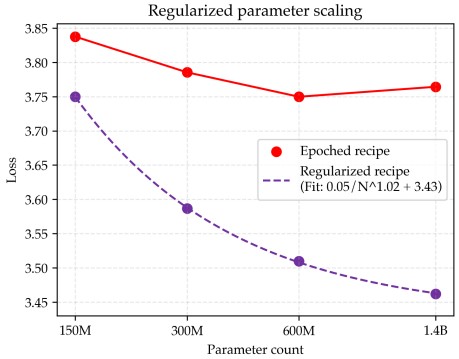

| Tuned $H$ | 150M | 300M | 600M | 1.4B |
|---|---|---|---|---|
| Learning rate | 3e-3 | 3e-3 | 1e-3 | 1e-3 |
| Epoch count | 16 | 16 | 8 | 8 |
| Weight decay | 0.8 | 1.6 | 3.2 | 3.2 |

Figure 3: **Power law scaling from jointly tuning regularization.** We compare the standard recipe in Figure 2 (red line) to our regularized recipe that jointly tunes learning rate, epoch count, and weight decay (purple line). After regularization, the loss decreases proportional to $\approx \frac{1}{N}$. The power law predicts that as $N \to \infty$, the best model achieves $3.43$ loss.

## 4 ENSEMBLE SCALING

The regularized recipe offers a straightforward way to improve performance by taking $N \to \infty$. Can different training algorithms better leverage the data under infinite compute? In this section, we consider ensembling (Dietterich, 2000): independently train $K$ models and average their logits for generation (Section 4.1). In Section 4.2, we show how ensembling can outperform parameter scaling at fixed parameter counts and under the limit as total parameter count approaches infinity.

---

[1]For this power law and all following ones, the units of parameters and tokens will be in billions for cleaner visualization and comparison. This only affects the numerator of the scaling law, not its asymptote or exponent.

[2]Because of run-to-run variance of the points forming the power law, we share a sensitivity analysis in Appendix I.1 showing that the asymptotes vary by at most 0.02 loss across 3 seeds.

In Section 4.3, we construct our best recipe composing regularized parameter scaling and ensemble scaling by taking the limit as both $N, K \to \infty$.

## 4.1 DEFINING ENSEMBLES

The ensembling pre-training algorithm $\mathcal{E}$ accepts a pre-training algorithm $\mathcal{A}$, trains $K$ members that are identical up to random seed $Z_i$ controlling data order and model initialization, and returns a model that averages the logits of the $K$ members. See Appendix D.1 for a full formal definition.

The number of FLOPs needed to generate from or evaluate an ensemble is simply the sum of the costs for all members. Since the number of FLOPs in a forward pass is approximately linear in parameter count (Kaplan et al., 2020; Hoffmann et al., 2022), we will consider an ensemble's total parameter count as $NK$ when comparing it to standard pre-training.

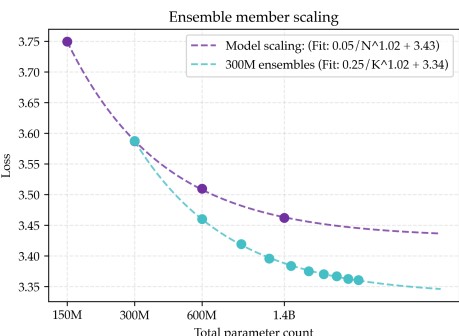

Figure 4: **Comparing scaling parameter count vs scaling ensemble member count.** Instead of scaling the parameter count of a single model, we can train an ensemble of smaller models and scale the number of ensemble members (resulting in $NK$ total parameters for $K$ ensemble members). Scaling up member count $K$ can similarly be fit by a power law with exponent approximately 1. Importantly, this law achieves a better asymptote than scaling $N$.

## 4.2 SCALING MEMBER COUNT INSTEAD OF PARAMETER COUNT

We compare the regularized and ensembling recipes under the best regularized hyperparameters from Section 3. In Figure 4, we find that the ensembling recipe's excess loss decreases close to a rate of $\frac{1}{K}$, similar to how the regularized recipe's excess loss decreases at a rate close to $\frac{1}{N}$. Under infinite compute, the ensembling recipe's ($N = 300$M, $K \to \infty$) asymptote is 3.34, which is lower than the regularized recipe's ($N \to \infty, K = 1$) asymptote of 3.43. Thus, for large $N$, it is better to train multiple small models instead of a single large model. In fact, even the $K = 3$ ensemble outperforms the regularized recipe's asymptote.

Why does ensembling improve over parameter scaling? Allen-Zhu and Li (2023) shows that ensembling helps when the data can be well-classified with one of many features but is best classified using all such features. Under this "multi-view" structure, they find that a single model is only learns one feature, whereas each ensemble member learns different features. [3]

## 4.3 JOINT SCALING RECIPE COMPOSING PARAMETER AND ENSEMBLE SCALING

Although the ensembling recipe outperforms parameter scaling, we can compose both by taking the number of members and the size of each member to infinity ($N, K \to \infty$). To estimate the best possible loss of a joint scaling recipe, we take two limits:

$$\hat{\mathcal{L}}_D = \lim_{N \to \infty} \lim_{K \to \infty} \min_H \mathcal{L}\left(\mathcal{E}_\mathcal{A}\left(D, N, K, H\right)\right)$$

As long as $\min_H \mathcal{L}\left(\mathcal{E}_\mathcal{A}\left(D, N, K, H\right)\right)$ monotonically decreases in $N$ and $K$ when fixing the other variable, the value does not depend on the order of the limits. We choose this order as it results in the most convenient hyperparameter tuning (Appendix D.6). For the inner limit, we cannot fully find locally optimal hyperparameters due to experimental constraints. Instead, we use the heuristic of taking the optimal regularized hyperparameters with $2\times$ epochs and $0.5\times$ weight decay (Appendix D.4).

In Figure 5, we show how we take this double limit. Our final estimate for the joint scaling recipe's loss is 3.17, which is much better than the regularized and unregularized losses of 3.43 and the 3.75.

---

[3]In Appendix D.2, we find optimally tuned ensembles match this intuition. Slightly overfitting each ensemble member beats an ensemble using the best regularized hyperparameters.

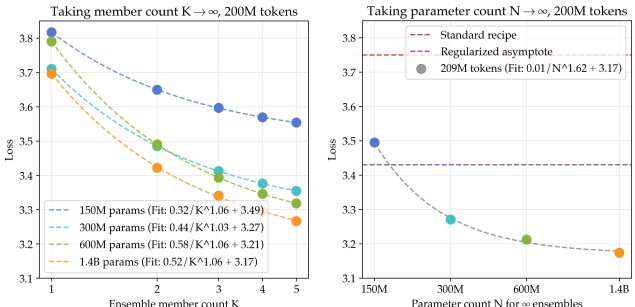

Figure 5: **Composing the regularized and ensembling recipes under the double limit.** Left: For each $N$, we fit a power law on the loss as $K$ increases. We select hyperparameters for low asymptotes instead of loss at small $K$. Right: We take the asymptotes from the left plot and fit a power law to capture how the asymptote changes for bigger ensemble members. This law's asymptote estimates the best possible loss under the joint scaling recipe.

## 5 SCALING THE SEED TOKEN COUNT UNDER INFINITE COMPUTE

Do our loss improvements at 200M tokens generalize to larger scales? In Sections 5.1 and 5.2, we first measure the best possible loss of our recipes at higher token counts up to 1.6B tokens. We contextualize the loss improvement and **data efficiency** of a recipe by interpolating how much data the standard recipe would need to match performance. In Section 5.3, we fit data-scaling laws to extrapolate how our recipes would perform at even higher token counts.

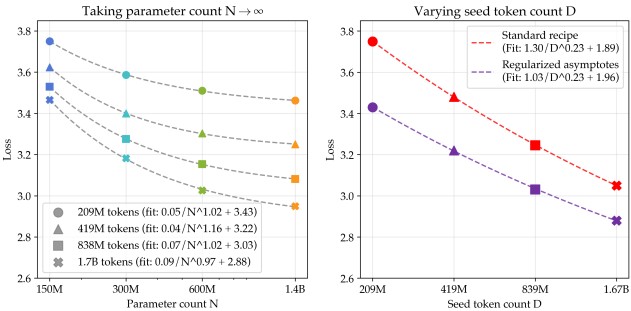

Figure 6: **Scaling the seed token count for single models.** We first consider the best loss of the standard recipe tuning epochs and parameters (red points, right). We then consider the best loss of the regularized recipe by fitting parameter scaling laws across four token counts (left) and taking their asymptotes (purple points, right). We fit data-scaling laws (red and purple lines) to extrapolate performance as seed token count increases.

### 5.1 DATA SCALING LAWS FOR SINGLE MODEL RECIPES

As shown in Section 2, the standard recipe overfits and does not admit a monotone scaling law. Instead, we search for the best parameter count and hyperparameters at each of our four data scales (Appendix E.1). Given these four estimates of the best loss at each token count (Figure 6, right), we fit a data scaling power law shown as the red line using $\hat{\mathcal{L}}_D := \frac{A}{D^\alpha} + E$.

We characterize the best possible loss of the regularized recipe by estimating $\lim_{N \to \infty} \min_H \mathcal{L}(\mathcal{A}(D, N, H))$ as shown in Section 3. Since we have to compute asymptotes to build the points for the data scaling law, we follow a two step procedure shown in Figure 6.

**Measuring data efficiency.** We measure the data efficiency between two recipes at a fixed token count $D$. We first compute the effective data $D'$ that $\mathcal{A}_1$ would need to match $\mathcal{A}_2$. After interpolating $D'$ via the data scaling law of $\mathcal{A}_1$, we report the data efficiency as $\frac{D'}{D}$. This metric characterizes the regularized recipe asymptote as $2.29\times$ more data efficient than the standard recipe at 200M tokens.

Even without any extrapolation of the asymptote, the best 1.4B model at 200M tokens is $2.09\times$ more data efficient than our baseline.

## 5.2 DATA SCALING LAWS FOR ENSEMBLES

We repeat the above procedure for ensembles by following Section 4.3 and estimating $\lim_{N\to\infty}\lim_{K\to\infty}\min_H \mathcal{L}\left(\mathcal{E}_\mathcal{A}\left(D,N,K,H\right)\right)$ for each seed token count $D$. We visualize the three step procedure in Figure 7. At 200M tokens, the asymptote of the joint scaling recipe is $5.17\times$ more data efficient than the standard recipe. Without taking asymptotes, our best ensemble of five 1.4B models is itself $3.75\times$ more data efficient.

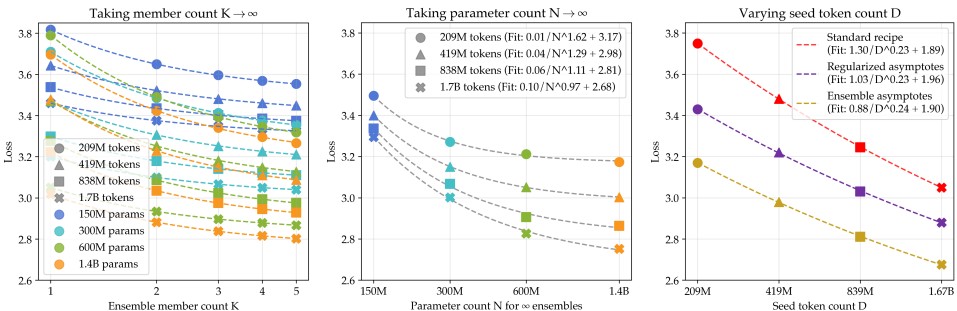

Figure 7: **Scaling the seed token count for ensembles.** Left: For fixed parameter and token count, we fit a power law in $K$, with hyperparameters optimized for the asymptote. Middle: We take the asymptote of the left 16 laws and fit a power law to measure how the asymptote changes in $N$. Right: We take the asymptote of the middle 4 laws and fit a power law to measure how the asymptote of asymptotes changes in $D$. At all token counts, we find over $2\times$ and $5\times$ data efficiency wins over the regularized and standard recipes respectively.

## 5.3 DATA SCALING ANALYSIS

Although the data scaling laws are expected to be noisy, they predict that all recipes decay at a similar rate with exponents between $0.23$ and $0.24$ and asymptotes between $1.89$ and $1.96$. Asymptotic statistics suggests that the asymptotes are equal if the algorithms achieve Bayes-optimal error under infinite data and compute, in which case their loss would be the entropy of text (Shannon, 1951; Van der Vaart, 2000). When the asymptote $E$ and exponent $\alpha$ of the laws are the same for two algorithms, there is a constant data efficiency improvement at all token counts determined by the numerators $A_1, A_2$, equal to $(A_2/A_1)^{\frac{1}{\alpha}}$. Our preliminary analysis suggests that our data efficiency wins will not disappear across all data scales even if they perform similarly under infinite data.

## 6 DATA EFFICIENCY UNDER PARAMETER CONSTRAINTS

The asymptotes of the regularized and ensembling recipes rely on arbitrarily high parameter models We study whether large models are necessary for data efficiency, either for the final model or for training. In Section 6.1, we distill an 8-ensemble of 300M members into a 300M student, preserving $83\%$ of the loss improvement with an $8\times$ smaller final model. In Section 6.2, we show self-distilling a 300M model into a student of the same size outperforms the teacher, removing the need for large parameter counts at training.

### 6.1 REDUCING FINAL PARAMETER COUNT VIA ENSEMBLE DISTILLATION

Even if our best scaling recipe helps in the limit as $N, K \to \infty$, can it help train models that are small relative to $D$? It is known that better large models can improve the performance of smaller models through knowledge distillation (Hinton et al., 2015; Yang et al., 2025; Team et al., 2025a). Since we are not bound by train compute, we can first pre-train a data-efficient teacher $M'$ on $D$ tokens using our existing recipes. Then, we sample from $M'$ unconditionally (i.e. with no prompt) to generate $D'$ tokens. We train our distilled student model $M$ from scratch on the mixture of $D$ and $D'$ (Kim and Rush, 2016).

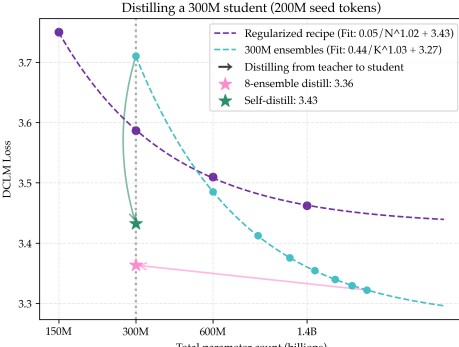

Figure 8: **Ensemble distillation and self-distillation.** We can compress our data efficiency gains into smaller models through distillation. Distilling an 8-ensemble teacher into a 300M student retains most of the loss improvement (pink star) and outperforms the regularized asymptote. Self-distillation with a 300M teacher and 300M student (green star) is surprisingly effective, matching the asymptote of the regularized recipe without increasing parameter count at training.

In Figure 8, we show the student model (pink star) obtained from using an 8-ensemble of 300M models (right-most blue point) with loss 3.32. Despite the $8\times$ smaller size, our distillation model attains a loss of 3.36, preserving $83\%$ of the ensemble improvement over the regularized 300M model loss of 3.57 (purple point). Our student outperforms the regularized recipe asymptote and matches the loss of a 4-ensemble (details in Appendix F).

## 6.2 REDUCING TRAIN PARAMETER COUNT VIA SELF-DISTILLATION

Is it possible to train a data-efficient model without high parameter count at train time as well? We consider this question for self-distillation where the teacher and student are of the same size and architecture. Many recent papers discuss how training a new student model on model generations can result in model collapse (Shumailov et al., 2024; Gerstgrasser et al., 2024; Dohmatob et al., 2024; Taori and Hashimoto, 2022).

On the contrary, by mixing together the $D$ real tokens and $D'$ synthetic tokens, we avoid collapse and can train a fresh student that vastly *outperforms* its teacher. In Figure 8, we show how using a 300M model as a teacher (blue point) results in a 300M student model (green star) that outperforms the best regularized 300M model (purple point). Why does self-distillation help?Allen-Zhu and Li (2023) provide theory interpreting self-distillation as implicitly ensembling the teacher and freshly initialized student.

## 7 DOWNSTREAM TASKS

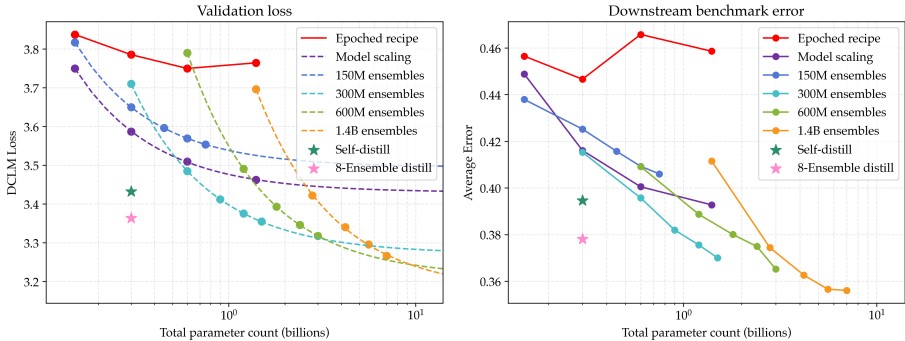

Figure 9: **Performance of pre-trained models on downstream tasks.** We have thus far been using validation loss (left) to seperate whether models are better pre-trained models or not. We evaluate the same models and ensembles on downstream benchmarks (right). Models with lower validation loss have lower average error across downstream benchmarks.

Although validation loss is known to correlate with capabilities of interest (Chen et al., 2025; Thrush et al., 2025; Gadre et al., 2024), we further test our models' general capabilities using downstream benchmarks. For evaluations that are informative for models at our scale, we take all of the accuracy-based benchmarks from Thrush et al. (2025), namely PIQA (Bisk et al., 2019), SciQ (Welbl et al., 2017), and ARC Easy (Clark et al., 2018). Notably, we did not evaluate on *any* benchmarks until

the end of the project after we selected the best recipes following validation loss, making these benchmarks a strong test of generalization.

In Figure 9, we show the validation loss (left) and downstream benchmark error (right) of our models for 200M tokens. Without regularization, the standard recipe does not benefit much from parameter scaling. Regularization (purple points) makes downstream accuracy scale smoothly with diminishing returns, similar to validation loss. Ensemble error mirrors loss with increasing $N$ and $K$ improving performance. Overall, our best ensemble outperforms our best unregularized model by over $9\%$ on average and our best distilled model outperforms the unregularized 300M model by $7\%$. See Appendix G for a full breakdown of results.

## 8    RELATED WORK

We cite additional related work on over-parametrized maching learning, distillation algorithms, synthetic data, and classical data-constrained deep learning in Appendix J.

**Scaling laws.**    Much of the success of language model pre-training was built upon scaling laws which accurately predict performance at a given resource budget (Hestness et al., 2017; 2019; Rosenfeld et al., 2019; Henighan et al., 2020; Kaplan et al., 2020; Sorscher et al., 2023; Hoffmann et al., 2022; Ruan et al., 2024; Cortes et al., 1993). Past work has studied scaling laws under constraints such as data and compute (Muennighoff et al., 2023; Goyal et al., 2024), hardware precision (Kumar et al., 2024), parameter count (Sardana et al., 2025; Springer et al., 2025; Gadre et al., 2024), and test-time compute (Brown et al., 2024; Snell et al., 2024). We show that past work (Muennighoff et al., 2023) does not account for over-fitting, fix this via regularization, and propose asymptote estimation as a new metric.

**Ensembling.**    Ensembling (Dietterich, 2000) is known to boost performance across settings including uncertainty estimation (Lakshminarayanan et al., 2017), image classification (Huang et al., 2017; Garipov et al., 2018), and reinforcement learning (van Hasselt et al., 2015). Deep ensembles are shown to follow power laws (Lobacheva et al., 2021) and not believed to outperform parameter scaling in certain theoretical models (Vyas et al., 2023; Ruben et al., 2024). We show how ensembling can be adopted for pre-training and build scaling laws to characterize loss. See Appendix D.5 for discussion on related alternatives.

**Distillation.**    Distillation spends compute to produce strong models with lower inference costs (Hinton et al., 2015) which we show with sequence knowledge distillation (Kim and Rush, 2016). For self-distillation, there is recent work showing how training on self-generated inputs can be harmful (Shumailov et al., 2024; Dohmatob et al., 2024; Taori and Hashimoto, 2022). Gerstgrasser et al. (2024) suggests that training on self-generated data can be helpful in limited scenarios, though their comparisons are neither compute-matched nor data-matched. The success of self-distillation aligns with prior evidence from data-constrained deep learning (Mobahi et al., 2020; Zhang et al., 2019). Notably, Allen-Zhu and Li (2023) show how self-distillation can be viewed as implicitly performing ensembling and distillation.

**Modern data-constrained pre-training.**    There are several recent works which study data-efficient pre-training and show the benefit of epoching (Muennighoff et al., 2023), rephrased synthetic data (Maini et al., 2024; Yang et al., 2024; DatologyAI et al., 2025; Ruan et al., 2024), diffusion language models (Prabhudesai et al., 2025; Ni et al., 2025), and energy-based models (Gladstone et al., 2025). These recent works do not aggressively regularize for optimal epoching nor build scaling laws to estimate infinite compute performance.

## 9    DISCUSSION

The success of classical ideas from data-constrained deep learning like regularization and ensembling suggests that there is free lunch on the table, encouraging us to revisit pre-training design decisions. We are also excited by methods that can better leverage extra compute for performance, in line with The Bitter Lesson (Sutton, 2019). We hope that evaluating scaling recipes via their asymptotes inspires more data-efficient algorithms for the future.

## 10  ACKNOWLEDGEMENTS

We thank Steven Cao, Sam Park, Jacob Mitchell Springer, Kaiyue Wen, Yu Sun, Nathan Hu, Meena Jagadeesan, Luke Bailey, Neil Band, Sally Zhu, Ben Spector, and Audrey Xie for their helpful discussions or feedback on the paper draft.

This work is a part of the Marin Project and the compute is supported by the Google TPU Research Cloud (TRC). TH was supported by a grant by HAI, DSO labs, gifts from Open Philanthropy, Amazon, Schmidt Sciences, the Tianqiao and Chrissy Chen Foundation and a grant under the NSF CAREER IIS-2338866, ONR N00014-24-1-2609, and DARPA Cooperative Agreement HR00112520013. PL was supported by DARPA Cooperative Agreement HR00112520013. This work does not necessarily reflect the position or policy of the government and no official endorsement should be inferred.

## 11  ETHICS

We hope that our work may be applied to settings beyond pre-training to improve data efficiency. We acknowledge our work may increase the amount of compute used for language model pre-training. We believe most other harms specific to our work apply to general language modeling research.

## 12  REPRODUCIBILITY

We open-source all of our runs on WandB and our code on Github.

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

CONTENTS

# A   CONTINUED PRE-TRAINING

We demonstrate the immediate applicability of our findings in settings outside of pre-training from scratch by improving the data efficiency of existing CPT recipes. We adopt the setup from Wang et al. (2025) of performing continued pre-training on Llama 3.2 3B Base (Grattafiori et al., 2024) with the `MegaMath-Web-Pro` mid-training dataset. To simulate a data-constrained setting, we restrict ourselves to only 4B seed tokens of the full 73B tokens. We evaluate math performance by via accuracy on a subset of representative benchmarks from Wang et al. (2025): GSM8K (Cobbe et al., 2021), MATH (Hendrycks et al., 2021), and MathQA (Amini et al., 2019).

In Table 1, we show that many of our results for pre-training from scratch transfer to continued pre-training setting. We start with a CPT baseline (Default) from the reference hyperparameters of Wang et al. (2025) which gives a $6.34\%$ improvement over the base model. We then apply our interventions from Appendix C.4 and Section 3 by decreasing batch size and epoching, and we find that weight decay was not helpful for CPT. Our final single model baseline provides an additional $5.23\%$ lift in average accuracy on top of the reference hyperparameters. See Appendix H for details.

We further show that ensembling eight epoched models provides an additional $4.76\%$ gain in average accuracy over just a single epoched CPT model ($K = 1$). We observe that average accuracy scales with increasing ensemble member count, and our best ensembles exceed the performance of a baseline continually pre-trained on the full 73B tokens, providing a $17.5\times$ data efficiency win.

Table 1: **Data efficiency improvements on OctoThinker.** We take reasoning mid-training data from Wang et al. (2025) and apply our data efficiency interventions of reducing batch size (Appendix C.4), epoching data, and ensembling multiple models. Our best ensemble utilizing 4B tokens outperforms vanilla CPT on 73B tokens following the original paper's training hyperparameters, resulting in a $17.5\times$ data efficiency improvement.

| Benchmarks | Llama 3B | CPT (4B tokens) | | | $K$-ensembles | | | CPT (73B tokens) |
|---|---|---|---|---|---|---|---|---|
| | | Default | Lower BS | Epoching ($K=1$) | $K=2$ | $K=4$ | $K=8$ | |
| GSM8K$_{(8\text{-}shot)}$ | 28.23 | 38.44 | 44.50 | 44.05 | 49.28 | 51.80 | **52.99** | 49.51 |
| MATH$_{(4\text{-}shot)}$ | 6.90 | 14.38 | 17.64 | 19.74 | 21.84 | 23.04 | **23.50** | 23.40 |
| MATHQA$_{(8\text{-}shot)}$ | 35.07 | 38.96 | 41.31 | 42.58 | 45.12 | **46.06** | 45.26 | 44.79 |
| Average | 24.25 | 30.59 | 34.48 | 35.82 | 38.79 | 40.35 | **40.58** | 39.23 |

# B   PROBLEM SETTING

**Pre-training algorithm.**   We instantiate the pre-training algorithm $\mathcal{A}$ using a standard pre-training recipe developed through the Marin project (`https://marin.community`) by following the best practices shared publicly and found internally.

- **Optimizer.** We train with AdamW, either with a default of $0.1$ or a tuned weight decay. We set other hyperparameters to standard defaults ($\beta_1 = 0.9, \beta_2 = 0.95, \epsilon = 10^{-8}$). We clip the norm of the gradient at $1$.

- **Learning rate.** We use a cosine learning rate schedule with a warmup for the first $1\%$ of training, decaying to 0 by the end of training. We always tune learning rate for all of our experiments. Every run has its own learning rate schedule and we never report the loss before the learning rate anneals to zero in the main body .

- **Architecture.** We train Llama-style auto-regressive language models. We specify the main architectural choices in Table 2. When scaling models, we change the initialization scheme to have variance inversely proportional to the hidden dimension (this is known to outperform $\mu$P (Yang et al., 2022) within our framework: `https://github.com/marin-community/marin/issues/621`. For other architectural choices, we default

to SiLU activations, untied word embeddings, and rotary position embeddings. We note that the non-standard scaling for the 1.4B model is a consequence of using presets in our pre-training framework.

- **Systems.** We train in mixed-precision with parameters in fp32 and compute + output in bf16. Most of our jobs were run on v4-64 or v4-128 TPUs, with bitwise-determinism for handling preemption and promoting reproducibility.

- **Data.** We pre-train using DCLM data (Li et al., 2025). We keep a fixed validation set of 1024 sequences (4 million tokens) across all experiments for clear comparison. When increasing the size of the train pool, we ensure that smaller pools are a subset of larger pools.

- **Data order.** We generate a random permutation of the windows after tokenization and use this same permutation across epochs. We note that performance might have been better if we used a unique permutation every epoch but keep this fixed across models which further reduces randomness of training.

| Parameter | 150M | 300M | 600M | 1.4B |
|---|---|---|---|---|
| Context Length | 4096 | 4096 | 4096 | 4096 |
| Hidden Dimension | 512 | 768 | 1024 | 2048 |
| Intermediate Dimension | 1792 | 2688 | 3584 | 7168 |
| Attention Heads | 8 | 12 | 16 | 16 |
| KV Heads | 8 | 12 | 8 | 8 |
| Layers | 6 | 12 | 24 | 16 |

Table 2: Model architecture configurations for different model sizes. We default to the 300M model if not specified.

## C  STANDARD PRE-TRAINING DETAILS

### C.1  LOCALLY OPTIMAL HYPERPARAMETERS

We are interested in finding the best setting of hyperparameters (e.g. learning rate, epoch count, and weight decay) for a fixed parameter count and token count in the data-efficient pre-training setting. To make this search problem tractable, we first discretize the space of hyperparameters (e.g. only try epoch counts that are powers of 2). Under this discretization, it is prohibitively expensive to try every possible hyperparameter selection within our grid. Therefore, we search for **locally-optimal hyperparameters** as defined below.

**Definition 1** (Locally-optimal hyperparameters)**.** *We define the neighborhood $\mathcal{B}(H)$ of hyperparameter tuple $H$ containing $m$ variables as the $2m$ neighbors from incrementing/decrementing exactly one of the variables. We say $H$ is locally optimal if and only if*

$$\forall H' \in \mathcal{B}(H), \ \mathcal{L}\left(\mathcal{A}\left(D, N, H\right)\right) \leq \mathcal{L}\left(\mathcal{A}\left(D, N, H'\right)\right)$$

Under certain assumptions, the locally-optimal hyperparameters are also globally optimal (for example, if the loss was convex in each dimension when the other variables are fixed). Though this may seem like a big assumption, we did not observe counter-examples to this in early experiments. One can verify this property for the single model losses presented in Figure 17.

Assuming this property, we use the following search procedure

1. Seed the search with initial runs around a best guess for optimal hyperparameters (heuristically set by us, using all runs until this point in time)

2. Take the best run so far and run its neighbors.

3. If any of its neighbors is better than the current candidate, the current candidate is suboptimal. Repeat step 2 with the new best run.

4. If this run is better than all of its neighbors, we consider it "certified" and terminate the search procedure.

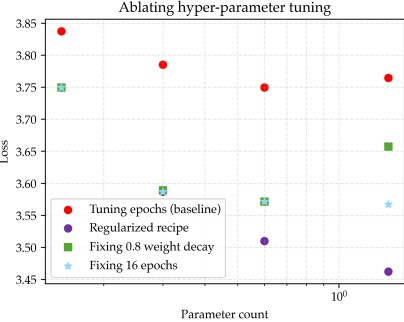

Figure 10: **Ablating joint tuning procedure.** We generally jointly tune learning rate, epoch count, and weight decay separately for each parameter count (purple). We show that trying to naively transfer hyperparameters across scales is a bad idea. Red: fixing weight decay to $0.1$ (default regularization). Green: assuming that the optimal weight decay for 150M models ($0.8$) is optimal across $N$. Blue: assuming that the optimal epoch count for 150M models ($16$) is optimal across $N$.

Though this procedure is quite expensive, we found it better than other natural heuristics which don't rigorously follow this procedure separately for each parameter and token count (Appendix C.2). Furthermore, it seems to give clean scaling in both loss and hyperparameters, suggesting that the hyperparameter optimization landscape is nice enough for this coordinate descent algorithm to work. We note that this is a simplified version of the assumptions used in Wen et al. (2025).

Our specific discretization was forcing learning rate to be $1$ or $3$ times a power of 10, epoch count to be an integer power of 2, and weight decay to be either $0.0$ or an integer power of 2 times $0.1$. To further restrict the search space, we set bounds based on initial experiments tuning these hyperparameters, with a maximum learning rate of 3e-3, maximum weight decay of $6.4$, and a maximum epoch count of $64$ (these bounds only triggered for three of our searches).

## C.2   ABLATING ON COORDINATE DESCENT

For the parameter scaling experiments, we jointly tune learning rate, epoch count, and weight decay. We find this joint tuning necessary for clean scaling. To demonstrate this, we consider three alternatives and show their scaling in Figure 10.

1. **Fixing weight decay** $0.1$ **and tuning learning rate and epoch count (red).** This baseline has already been shown to fail in Section 2.1.

2. **Jointly tuning weight decay only for 150M (green).** Here, we jointly tune the weight decay for the 150M model, finding the optimal value is $0.8$. We then assume this is the optimal value for higher parameter counts and correspondingly tune learning rate and epoch count. We find that this scaling is not even monotonic.

3. **Jointly tuning epoch count only for 150M (blue).** Here, we jointly tune the epoch count for the 150M model, finding the optimal value is $16$. We then assume this is the optimal value for higher parameter counts and correspondingly tune learning rate and weight decay. We find that this scaling plateaus much faster than the regularized recipe.

This shows the importance of jointly tuning both weight decay and epoch count at each model scale instead of blindly hoping for transfer.

## C.3   TUNED HYPERPARAMETERS

In Figure 11, we share the locally optimal hyperparameters we found for 4 different token counts and 4 different parameter counts. We notice a few trends when looking at the resulting hyperparameters and power laws.

- The optimal learning rate decreases for larger models, noted by prior work (Yang et al., 2022; Everett et al., 2024). The optimal learning rate does not strongly depend on the number of tokens.

- The optimal weight decay increases for larger models. Similarly, the optimal weight decay decreases for larger token counts. When fixing the parameter-to-token ratio, the weight decay stays around $0.8$.

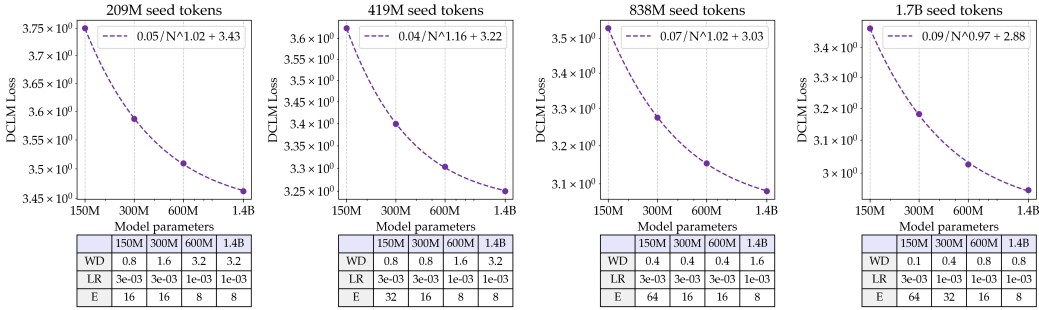

Figure 11: **Tuned hyperparameters for regularized scaling.** We show the optimal hyperparameters tuned seperately for each parameter and token count. We find that as parameter count increases, optimal weight decay goes up, optimal epoch count goes down, and optimal learning rate goes down. We find the trends for weight decay and epoch count hold when token count decreases.

- The optimal epoch count decreases for larger models. Similarly, the optimal epoch count increases for larger token counts. When fixing the parameter-to-token ratio, the epoch count stays around 16.
- The power laws fit across all token counts share similar scaling exponents close to 1. This holds even though almost every model is over-parameterized at 200M tokens and under-parameterized at 1.6B tokens.

### C.4 HYPERPARAMETER ABLATIONS

We perform additional ablations on hyperparameters to understand their role beyond their optimal values. We start from a recipe of single epoch pre-training with 0.1 weight decay and tuned learning rate, and build our way up to tuning all hyperparameters.

**Batch size.** It is known that when optimizing for throughput, it is best to train at the "critical batch size" to best utilizes hardware (McCandlish et al., 2018). However if we drop this constraint and instead measure performance for a fixed number of data points, we find that it is best to use smaller batch sizes, as shown in Figure 12, left, corroborating prior work in optimization (Smith et al., 2020; Keskar et al., 2017; LeCun et al., 1998; Marek et al., 2025). We use a batch size of 64, which is the smallest size that is practical for our hardware.

**Weight decay.** It is known that regularization can further improve generalization when repeating data (D'Angelo et al., 2024). Figure 12, right shows how varying the weight decay impacts epoched models (1.4B parameters for 8 epochs vs 300M parameters for 16 epochs). More over-parametrized models require a larger amount of regularization (3.2 vs 1.6). Without optimally tuning weight decay, one may draw the incorrect conclusion that larger models are worse than smaller models in the data-constrained setting. We also reproduce findings that higher weight decay enables a higher optimal learning rate and epoch count, shown by contrasing Figure 2 and Figure 3.

We find that increasing weight decay strongly changes the training dynamics. Though the train and validation losses decrease much slower at the start of training with high weight decay, they decrease rapidly by the end of training, eventually beating the run with low weight decay. In Figure 13, we visualize this for the best run with weight decay 0.1 and the best run tuning weight decay. This phenomenon also holds when using a higher weight decay on top of the best 0.1 weight decay hyperparameters. This is in line with previous findings in optimization research (Wen et al., 2025; Liu et al., 2025; D'Angelo et al., 2024) and suggests that we should only look at the performance at the end of training.

### C.5 OVERFITTING ANALYSIS

In Section 2.1, we discuss how introducing too many epochs or parameters results in validation loss going up which we believe is due to over-fitting. In Figure 14, we track train loss for the interventions

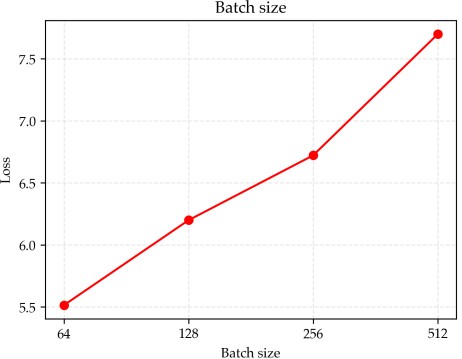
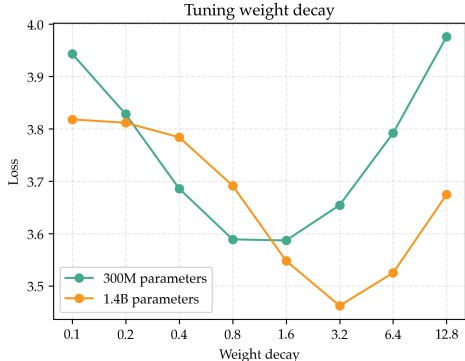

Figure 12: **Re-evaluating hyperparameters for standard pre-training.** Left: smaller batch sizes are better, we stop at 64 since this is the smallest our hardware practically supports (shown for 1 epoch training with 0.1 weight decay and a fixed learning rate of 3e-3). Right: weight decay helps, and the optimal weight decay is higher for larger models (300M is 16 epochs 3e-3 learning rate, 1.4B is 8 epochs 1e-3 learning rate).

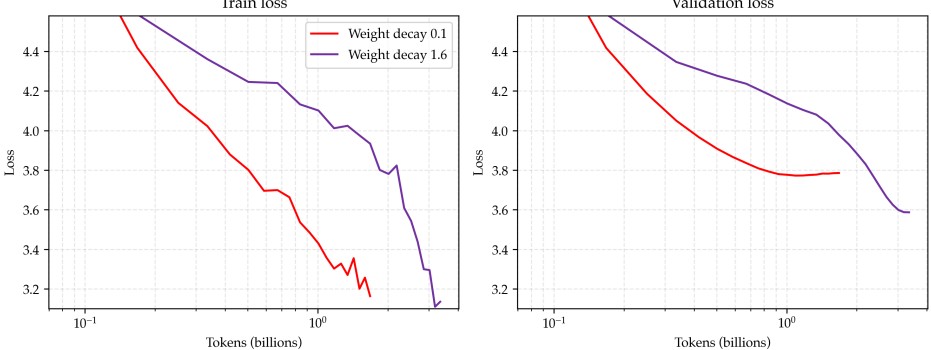

Figure 13: **Loss trajectories for different weight decays.** We compare the best run with default weight decay (8 epochs, 1e-3 learning rate, 0.1 weight decay) and the best run with tuned weight decay (16 epochs, 3e-3 learning rate, 1.6 weight decay) for 200M tokens and 300M parameters. We find that loss for runs with high weight decay decreases much more slowly at the start of training, but quickly decreases near the end of training.

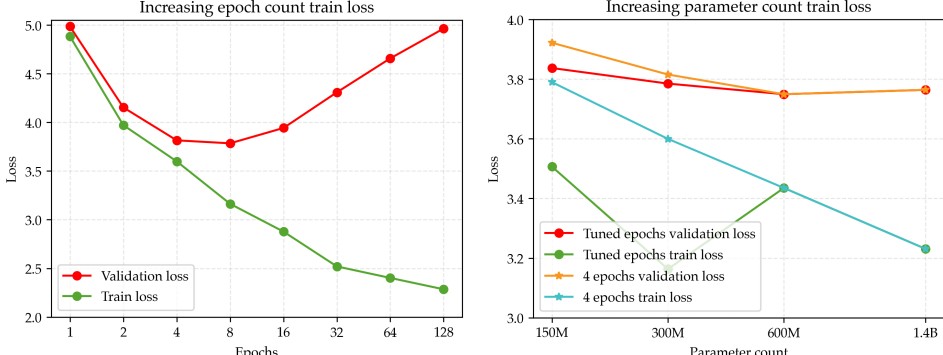

Figure 14: **Train losses for epoching and parameter scaling.** Left: Increasing epoch count results in train loss decreasing while validation loss starts increasing. Right: Increasing parameter count does not always decrease loss, potentially due to optimal epoch count changing (8, 8, 4, 4). Train loss monotonically goes down when restricted to 4 epochs.

of increasing epoch count and parameter count. On the left, we show how increasing epoch count monotonically decreases train loss but eventually results in validation loss going up. On the right, we show how increasing parameter count results in erratic changes in train loss. We hypothesize this is because the optimal epoch count changes from 8 for the first two models to 4 for the last two models. We find that when we restrict all models to only use 4 epochs, train loss decreases monotonically and validation loss still goes up, suggesting over-fitting.

We note that another reason over-fitting may be happening is because our 1.4B model trades depth for width. We did not recognize our model scaling was non-standard until the majority of our experiments had finished because these were the default settings in our pre-training framework. We do not think this is a severe issue since correctly tuning weight decay seems to correct for the fact that this architecture has less layers. Moreover, the large improvement from weight decay is also suggestive of the fact that larger models are over-fitting.

# D  ENSEMBLING DETAILS

## D.1  ENSEMBLING FORMALIZATION

The ensembling pre-training algorithm $\mathcal{E}$ accepts a standard pre-training algorithm $\mathcal{A}$ and trains $K$ members that are identical except for random seed $Z_i$ controlling the data order and model initialization. The output of the ensembling algorithm is a model that averages the logits of the $K$ models, computed by querying all $K$ models. More formally, we define the ensembling algorithm as

$$\mathcal{E}_{\mathcal{A}}(D, N, K, H) = \text{LogitAvg}\left(\{\mathcal{A}\left(D, N, Z_i, H\right)\}_{i \in [K]}\right)$$

for randomness $Z_i$, where $\text{LogitAvg}$ produces a model with likelihood for a sequence $x$ given by

$$\text{LogitAvg}\left(M_{i \in [K]}\right)(x) \propto \exp\left(\frac{1}{K} \sum_{i \in [K]} \log\left(M(x)\right)\right)$$

## D.2  OVERVIEW OF ENSEMBLING TUNING

Given the success of ensembles, we study how to tune their hyperparameters. In Figure 4, the ensemble members were trained using the best hyperparameters for a single model, but this may not be optimal for large $K$ ensembles. More formally,

$$\underbrace{\operatorname*{argmin}_{H} \mathcal{L}\left(\mathcal{A}(D, N, H)\right) = \operatorname*{argmin}_{H} \mathcal{L}\left(\mathcal{E}_{\mathcal{A}}(D, N, K=1, H)\right)}_{\text{best for single model}} \neq \underbrace{\operatorname*{argmin}_{H} \lim_{K \to \infty} \mathcal{L}\left(\mathcal{E}_{\mathcal{A}}(D, N, K, H)\right)}_{\text{best for ensemble member asymptote}}$$

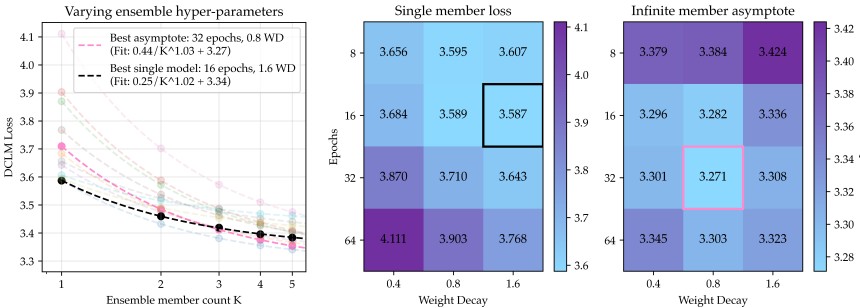

Figure 15: **Tuning hyperparameters of ensemble members for lowest asymptote under $K \to \infty$.** We construct ensembles for different $K$ when varying epoch count and weight decay. We find that the ranking between hyperparameters changes across $K$ (left) and that the infinite member asymptote benefiting from more epochs and less weight decay per member.

In Figure 15, we train ensembles for $K \in [5]$ as we vary weight decay and epoch count[4]. To estimate the best hyperparameters as $K \to \infty$, we fit a power law and refer to the asymptote. We find that the ranking between hyperparameters changes depending on $K$ (Figure 15, left). Importantly, the optimal hyperparameters for $K = 1$ (black) are not the best asymptote under $K \to \infty$ (pink), which benefits from more epochs and less weight decay. This intuitively corresponds to each member being more overfit to the data, which might be helpful to learn different views of the data. Selecting the hyperparameters when considering $K \to \infty$ improves the ensembling asymptote from 3.34 to 3.27 (Figure 15, right).

Another design choice for our ensembles is the source of randomness $Z_i$. Though we vary both the data order and model initialization, we find that either one by itself obtains most of the benefits of ensembling and reserve a full analysis to Appendix D.3.

### D.3    SEED SCIENCE

For our training runs, the randomness only comes from the sampled initialization (train seed) and shuffled data order (data seed). We first characterize the run-to-run variance when varying both seeds, only train seed, or only data seed. We train 5 models for each of the three randomness options. When using the optimal hyperparameters for a 300M model with 200M tokens, the standard deviation is estimated to be 0.008207 for both seeds, 0.007605 for only train seed, and 0.007213 for only data seed. This is in line with prior work that shows how only a small amount of instability is needed to induce the majority of the variance over a final model's loss (Summers and Dinneen, 2021; Jordan, 2024).

We now measure the loss of ensembles with these sources of variance in Figure 16. Either of these sources delivers most of the benefit of ensembling, with data order helping more. Considering the marginal benefit of introducing additional sources of randomness, we didn't strongly explore adding more sources of randomness during training.

### D.4    HYPERPARAMETER TUNING FOR ENSEMBLES

Similar to parameter scaling, we hope to find the locally optimal hyperparameters for a given parameter count and token count. However, we care about the hyperparameters as $K \to \infty$, not at $K = 1$ (as discussed in Section 4.2). Since it is too experimentally prohibitive to search for locally optimal hyperparameters for the asymptote, we study how the hyperparameters change relative to the optimal $H$ for single models.

We repeat the analysis in Figure 15 for 3 different parameter counts as well as 2 different learning rates at 300M parameters, shown in Figure 17. We find that for all of the displayed settings, our best run comes from an ensemble with the same learning rate, half weight decay, and double epoch count relative to the optimal single model hyperparameters. For the three parameter counts we display, we

---

[4]We find that optimal learning rate is generally consistent across $K$, Appendix D.4

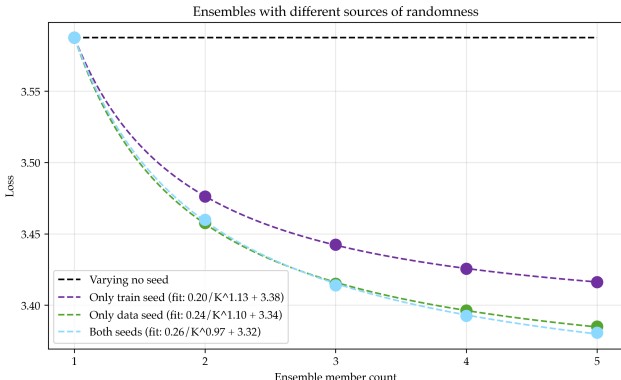

Figure 16: **Sources of randomness for ensembling.** Only varying train or data seed is enough to induce the benefits of ensembling. Varying data seed (i.e. order) is better between these two.

verify that these are locally optimal hyperparameters. In fact, across all of the ensembles we trained across our scales (including many that are not directly visualized), we find only one counter-example to this heuristic. This occurs for 1.4B models trained on 200M tokens, where the hyperparameters that minimize the asymptote do not halve the weight decay. We suspect this change occurs because this is our most over-parameterized setting, and our scaling laws use this best run over the heuristic for this single setting.

### D.5    ALTERNATIVES TO ENSEMBLING

The success of ensembling suggests alternative parameterizations that might also boost data efficiency. We discuss commonly considered ones here, which we were not able to get to outperform ensembling.

#### D.5.1    MIXTURE-OF-EXPERTS

Ensembling may qualitatively seem similar to training with Mixture-of-Experts (MoE). However, we find an important distinction: when training an ensemble, the learning trajectory for each model is completely independent of each other, whereas for a MoE, it is still a single learning trajectory. Unfortunately, the intuition from Allen-Zhu and Li (2023) suggests that the sparsity of the MoE architecture is not guaranteed to benefit from "multi-view" data in the way ensembles do. In their paper, they consider a simplified analogue where they construct a model that runs the ten models in parallel and takes the gradient step through the ensemble jointly. They find that this barely improves performance over a single model. In early experiments, we were able to reproduce this phenomenon, with a jointly trained 10 ensemble of models outperforming a single model by only 0.02 loss. Therefore, we hypothesize that if MoE's were to help, their benefits would come from the drop-out aspect instead of the sparsity aspect, which does not require MoE's (note that we did not tune or consider drop-out in this work, though we expect it to further help performance). We hope future experiments can settle this intuition more concretely.

#### D.5.2    MODEL SOUPS

Prior works have shown that averaging the weights of independent training runs can result in better models (Wortsman et al., 2022). However, we note that most success from averaging weights comes at fine-tuning, not pre-training. We replicate these results in our own settings, with model soups achieving close to random performance on downstream benchmarks (Table 5) but slightly outperforming ensembles in continued pre-training (Table 7).

One intuition for this discrepancy is that models need to be in the same "loss basin" for averaging to help final performance, and pre-trained models enter different loss basins (Singh and Jaggi, 2023; Ainsworth et al., 2023). Past studies also design compute-efficient algorithms for merging models trained from scratch, but they find that the more expensive procedure of distillation outperforms

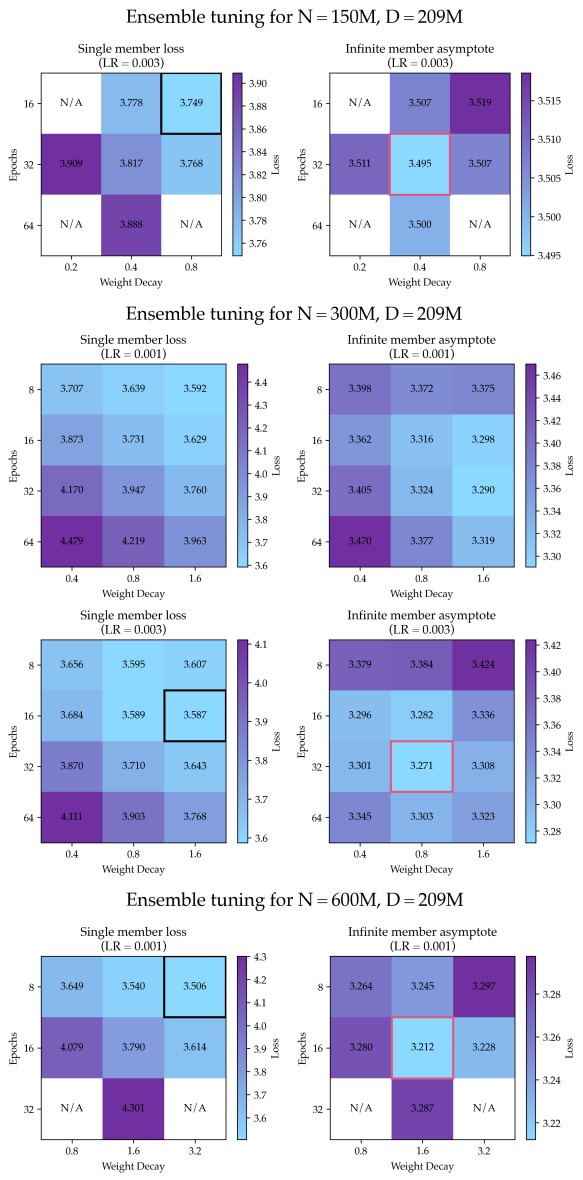

Figure 17: **Single model and asymptote loss when varying epoch count and weight decay for different model sizes, token counts, and learning rates.** We display an extended version of Figure 15 for 200M tokens with 150M, 300M (sub-optimal and optimal LR), and 600M parameter models. For these settings, the "double epoch, half weight decay" heuristic correctly predicts the best ensemble. This heuristic is consistent with all of our parameter and token counts except for our most over-parameterized setting.

their method (Singh and Jaggi, 2023). Since we are in the infinite compute regime, we opt to use distillation over model merging for the best performance.

## D.6 ORDER OF LIMITS

In Section 4.3, we are interested in computing the best possible performance of ensembles as $N$ and $K$ both go to $\infty$. There are a couple of different ways to compute this, some of which are enumerated below

**Double Limit 1 (Our Approach).**    We first solve for the limit as $K \to \infty$ by tuning asymptotes and then solve for the outer limit via a second power law over the inner asymptotes.

$$\lim_{N \to \infty} \lim_{K \to \infty} \min_{H} \mathcal{L}\left(\mathcal{E}_{\mathcal{A}}\left(D, N, K, H\right)\right)$$

**Hypothetical Double Limit 2.**    We can flip the above order and instead take $N \to \infty$ before $K \to \infty$, corresponding to

$$\lim_{K \to \infty} \lim_{N \to \infty} \min_{H} \mathcal{L}\left(\mathcal{E}_{\mathcal{A}}\left(D, N, K, H\right)\right)$$

**Hypothetical Double Limit 3.**    Following literature in compute-optimal scaling, we can find the best possible performance for a given compute budget $C$ and take $C \to \infty$.

$$\lim_{C \to \infty} \min_{\substack{H, N, K \\ \text{s.t. FLOPs}(D,N,K,H)=C}} \mathcal{L}\left(\mathcal{E}_{\mathcal{A}}\left(D, N, K, H\right)\right)$$

We believe that our approach is experimentally much more convenient than the hypothetical approaches even though they are equivalent in output under assumptions. We share our reasoning by answering the following questions comparing the approaches.

The core assumption that we will make is that $f(N, K) = \min_{H} \mathcal{L}\left(\mathcal{E}_{\mathcal{A}}\left(D, N, K, H\right)\right)$ is monotone in $N$ and $K$ when the other is fixed. Across all of our experiments, we do not observe any contradictions to this assumption as long as we tune regularization (for examples, refer to Figure 6 and Figure 7).

- **Are Double Limit 1 and 2 mathematically equivalent?** Mathematically, both limits are equivalent. For a quick proof, define $k_i := \lim_{K \to \infty} f(D, N)$ and $n_i := \lim_{N \to \infty} f(D, N)$. By monotonicity, both $k_i$ and $n_i$ exist and are non-increasing sequences. Define $k = \lim_{N \to \infty} k_i$ and $n = \lim_{K \to \infty} n_i$ which exist for the same reason. Note that $f(N, K) \le n_i$, and since limits preserve inequality, it follows that $k_i \le n$, and further follows that $k \le n$. By repeating this argument in the reverse direction, it follows that $k = n$.

- **Are Double Limits 1 and 2 mathematically equivalent to 3?** If we make the same monotonicity assumption as earlier, then we have that the minimization problem is monotonic in $C$. With this, we can apply a similar argument as above to show that the limit converges and is equivalent to $k$ and $n$.

- **How do we solve for the inner limit in Double Limit 1?** It is computationally prohibitive to tune all the hyperparameters for each choice of $N$ and $K$. Therefore, we would prefer searching for optimal hyperparameters once per choice of $N$ to fit the outer limit. Since we are only interested in the performance as $K \to \infty$ increases, we can reasonably approximate this via our hyperparameter heuristic from Appendix D.4 without ever determining the optimal hyperparameters at lower values of $K$.

- **Why do we prefer Double Limit 1 to Double Limit 2?** We first make the observation that tuning hyperparameters depends on $N$, but if we follow the previous bullet's asymptote tuning heuristic, tuning hyperparameters does not depend on any finite value of $K$. Therefore, Approach 2 would still have to fit hyperparameters for each $N, K$, whereas Approach 1 avoids this by fitting it once per $N$.

- **Why do we not use Double Limit 3?** In Double Limit 1, we keep the hyperparameter as the scaling axis, instead of Double Limit 3 which sets compute as the scaling axis. When we

choose to scale the hyperparameter, we can use our locally optimal hyperparameter search algorithm to find the best possible performance for that hyperparameter. This is difficult when scaling compute, since our hyperparameters such as model size and epoch count influence the compute spent during pre-training. Our preference reflects how practitioners typically use Chinchilla Approach 3 (fitting loss for best run given $N$, $D$, closer to Double Limit 1) over Chinchilla Approach 1 (fitting the envelope of the runs, closer to Double Limit 3) (Besiroglu et al., 2024).

# E  DATA SCALING

## E.1  EPOCH TUNED BASELINE

Unlike regularized parameter scaling and ensemble scaling where we estimate the best possible loss via asymptotes (Section 5), epoch tuning eventually over-fits. To estimate the best possible loss under epoch tuning for each token count, we use the following procedure.

1. For a fixed token count $D$ and fixed parameter count $N$, search for locally optimal hyperparameters while fixing weight decay to be $0.1$
2. Perform this for all parameter counts $N$ and token counts $D$

After following this procedure, we found that 600M models and 1.4B models were within $\approx 0.02$ loss of each other and were much better than the other models, as shown in Figure 18. Across all our token scales, 600M models slightly outperformed 1.4B models, which we discuss in C.5. Therefore, we take the 600M performance as an estimate of the best possible loss under regularized parameter scaling. We believe the performance of this algorithm would be better under better width/depth/architectures, though we expect this benefit to translate to our other recipes as well.

## E.2  SCALING PARAMETER COUNT

For each token count $D$, we characterize the best possible loss of the regularized recipe by estimating $\lim_{N\to\infty} \min_H \mathcal{L}\left(\mathcal{A}\left(D, N, H\right)\right)$ via the asymptote of the power law as described in Section 3. Since we now have to compute asymptotes to build the points for the data scaling law, we follow a two step procedure as visualized in Figure 6.

1. **Varying parameter count (left plot).** We first follow the locally-optimal hyperparameter search detailed in Section 3 to find the best possible models for four parameter counts across four seed token counts. These are the 16 points in Figure 6, left.
2. **Varying token count (right plot).** For each seed token count, we can fit a power law and use the asymptote $E_D$ as an estimate of $\mathcal{L}_D^*$. These four $(D, E_D)$ tuples form the purple points in Figure 6, right. We then fit a data scaling power law over these asymptotes, shown as the purple line.

## E.3  SCALING MEMBER AND PARAMETER COUNT

Following Section 4.3, we repeat the above procedure for ensembles by taking $\lim_{N\to\infty} \lim_{K\to\infty} \min_H \mathcal{L}\left(\mathcal{E}_\mathcal{A}\left(D, N, K, H\right)\right)$ for each seed token count $D$. This results in the following three step procedure, visualized in Figure 7.

1. **Varying ensemble member count (left plot).** In Figure 7, left, we picture the losses of ensembles across our 4 token counts, 4 parameter counts, and 5 member counts with hyperparameters selected for better asymptotes. For each of the 16 pairs of $D$ and $N$, we use a power law to extrapolate the performance as we take member count $K \to \infty$.
2. **Varying member parameter count (middle plot).** The asymptotes of the above 16 limits are visualized in Figure 7, middle. Given these asymptotes, we can fit a second set of 4 power laws that extrapolate the loss as we take parameter count $N \to \infty$.
3. **Varying token count (right plot).** The asymptotes of these 4 limits are visualized by the gold points on Figure 7, right. We then fit a data scaling law, depicted by the gold line.

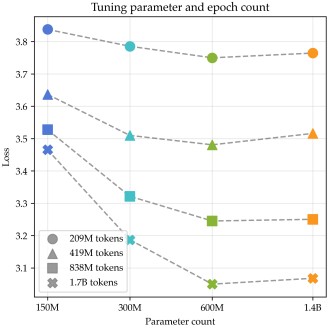

Figure 18: **Scaling seed token count for epoching and parameter scaling.** For each seed token count, we train models of varying $N$ with jointly tuned learning rate and epoch count with a weight decay of $0.1$. We take the best possible model at each token count as an estimate of the best performance under the standard recipe.

## F    DISTILLATION DETAILS

### F.1    DATA GENERATION

For all distillation experiments, we generate teacher data from the same model family: $K$-ensembles of 300M models with optimal hyperparameters for asymptotic performance. We choose to perform self-distillation with a 1-ensemble from this family rather than the 300M model from the regularized recipe to cleanly isolate the effect of distilling from a stronger teacher. In practice, we don't observe a significant difference: self-distillation from the 1-ensemble (blue point, Figure 8) gives a loss of 3.43 while self-distillation from the 300M regularized recipe (purple point, Figure 8) gives a loss of 3.44.

Because we are unconstrained by train compute, optimal distillation should never epoch on teacher data and instead generate more. We pre-generate a large pool of teacher distillation data by sampling unconditionally with temperature 1 using a high-throughput inference engine designed for batched workloads (Juravsky et al., 2025). For generating ensemble teacher distillation data, we experiment with inferencing both the logit averaged ensemble as well as the individual members. We observe better student performance using the individual members.

### F.2    HYPERPARAMETERS

For both ensemble distillation and self-distillation, we search for optimal hyperparameters using a procedure similar to Appendix C.1. Our distillation recipe also introduces a new hyperparameter which we refer to as the mixing ratio: the ratio of batches of real data to synthetic data. A mixing ratio of $1 : 1$ indicates that we take the same number of gradient steps on real data as teacher-generated data. For example, if we have 209M tokens of real data that we wish to epoch on 16 times, a $1 : 1$ mixing ratio would require $209 \cdot 10^6 \times 16 = 3.3\text{B}$ tokens of teacher-generated distillation data. We find tuning the mixing ratio to be important for performance.

We detail the exact values of hyperparameters in Table 3. Interestingly, we observe that optimal weight decay for distillation is lower than that of our regularized recipes, in line with standard practice. In addition, we find that ensemble distillation admits a higher optimal mixing ratio, likely due to the greater diversity from the teacher's synthetic data. Our ensemble distillation run trains on a total of $16 \times 209 \cdot 10^6 \times (1 + 9) = 33.4\text{B}$ tokens, while our self-distillation run trains on a total of $16 \times 209 \cdot 10^6 \times (1 + 3) = 13.4\text{B}$ tokens. Due to limitations in inferencing, we only generate 10B tokens each of ensemble distillation and self-distillation data, so our ensemble distillation may epoch up to 3 times on the teacher data.

| Parameter | Ensemble Distill | Self-Distill |
| --- | --- | --- |
| Learning Rate | 3e-3 | 3e-3 |
| Weight Decay | 0.1 | 0.1 |
| Mixing Ratio | $1 : 9$ | $1 : 3$ |
| Epochs | 16 | 16 |

Table 3: Optimal hyperparameters for ensemble and self-distillation.

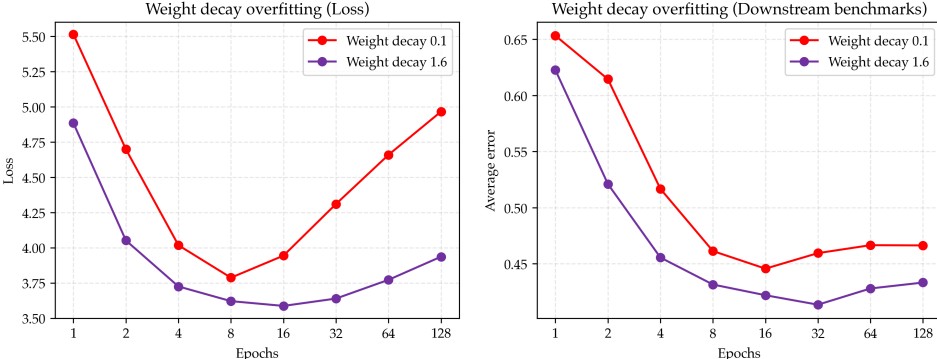

Figure 19: **Effect of regularization on overfitting for downstream benchmarks.** Downstream benchmarks also reflect the benefit of heavy regularization on performance. The effect of overfitting on downstream benchmarks (right) appears at twice the epoch count compared to validation loss (left).

### F.3 MIXING DATA ABLATION

We provide a token-matched ablation for the effect of mixing in the real pre-training data when doing self-distillation. As in Appendix F.2, we start with the same pool of 10B pre-generated tokens from a 300M 1-ensemble. Perfect distillation into a student model of the same size (with an infinite amount of teacher data) would achieve the same loss as the teacher.

We compare self-distillation with and without mixing in real data. For mixing in real data, we epoch the real data 16 times and use a $1 : 1$ mixing ratio so that the total number of tokens we train on is less than 10B. For no mixing, we simply train on a subset of the pre-generated pool. Both methods use the same learning rate and batch size, train on a total of $16 \times 209 \cdot 10^6 \times (1 + 1) = 6.688$B tokens, and never repeat the synthetic teacher data. For no mixing, we additionally search over weight decay.

Table 4 shows that without mixing in real pre-training data, self-distillation is substantially worse than the teacher model (as one might expect). Mixing data allows for self-distillation to exceed the teacher model.

|  | Teacher Model | Self-Distill ($1 : 1$ mixing) | Self-Distill (No mixing) |
|---|---|---|---|
| Val Loss | 3.7103 | 3.4373 | 4.0693 |

Table 4: Effect of mixing real pre-training data for self-distillation.

## G DOWNSTREAM TASK DETAILS

### G.1 DOWNSTREAM TASKS

We provide a full breakdown of downstream benchmark scores per model type in Table 5. We use `lm-evaluation-harness` (Gao et al., 2024) for our evaluations.

### G.2 HYPERPARAMETER TUNING

We find that hyperparameter tuning from validation loss transfers to downstream benchmarks as well. Figure 19 (left) shows how adding heavy regularization with weight decay (with a fixed learning rate of 3e-3) shifts the overfitting point based on validation loss to the right and down. We observe a similar effect in Figure 19 (right), although the overfitting threshold (in epochs) is twice the threshold observed for validation loss.

| Model type | | ARC-Easy (%) | PIQA (%) | SciQ (%) | Avg (%) |
|---|---|---|---|---|---|
| Unregularized model scaling | 150M | $40.95_{\pm 1.01}$ | $59.68_{\pm 1.14}$ | $62.40_{\pm 1.53}$ | $54.35_{\pm 0.72}$ |
| | 300M | $41.96_{\pm 1.01}$ | $61.15_{\pm 1.14}$ | $62.90_{\pm 1.53}$ | $55.34_{\pm 0.72}$ |
| | 600M | $39.86_{\pm 1.00}$ | $59.90_{\pm 1.14}$ | $60.50_{\pm 1.55}$ | $53.42_{\pm 0.72}$ |
| | 1.4B | $40.61_{\pm 1.01}$ | $60.39_{\pm 1.14}$ | $61.40_{\pm 1.54}$ | $54.14_{\pm 0.72}$ |
| Model scaling | 150M | $41.29_{\pm 1.01}$ | $60.17_{\pm 1.14}$ | $63.90_{\pm 1.52}$ | $55.12_{\pm 0.72}$ |
| | 300M | $44.28_{\pm 1.02}$ | $61.81_{\pm 1.13}$ | $69.10_{\pm 1.46}$ | $58.39_{\pm 0.70}$ |
| | 600M | $47.10_{\pm 1.02}$ | $63.06_{\pm 1.13}$ | $69.70_{\pm 1.45}$ | $59.95_{\pm 0.70}$ |
| | 1.4B | $45.66_{\pm 1.02}$ | $63.82_{\pm 1.12}$ | $72.70_{\pm 1.41}$ | $60.73_{\pm 0.69}$ |
| 150M ensembles | $K = 1$ | $42.85_{\pm 1.02}$ | $60.88_{\pm 1.14}$ | $64.90_{\pm 1.51}$ | $56.21_{\pm 0.72}$ |
| | $K = 2$ | $44.61_{\pm 1.02}$ | $62.02_{\pm 1.13}$ | $65.80_{\pm 1.50}$ | $57.48_{\pm 0.71}$ |
| | $K = 3$ | $44.91_{\pm 1.02}$ | $62.08_{\pm 1.13}$ | $68.30_{\pm 1.47}$ | $58.43_{\pm 0.71}$ |
| | $K = 4$ | $45.83_{\pm 1.02}$ | $62.19_{\pm 1.13}$ | $69.20_{\pm 1.46}$ | $59.07_{\pm 0.70}$ |
| | $K = 5$ | $45.50_{\pm 1.02}$ | $62.30_{\pm 1.13}$ | $70.40_{\pm 1.44}$ | $59.40_{\pm 0.70}$ |
| 300M ensembles | $K = 1$ | $44.36_{\pm 1.02}$ | $62.95_{\pm 1.13}$ | $68.10_{\pm 1.47}$ | $58.47_{\pm 0.71}$ |
| | $K = 2$ | $46.72_{\pm 1.02}$ | $63.87_{\pm 1.12}$ | $70.70_{\pm 1.44}$ | $60.43_{\pm 0.70}$ |
| | $K = 3$ | $47.77_{\pm 1.02}$ | $64.74_{\pm 1.11}$ | $72.90_{\pm 1.41}$ | $61.80_{\pm 0.69}$ |
| | $K = 4$ | $48.36_{\pm 1.03}$ | $65.89_{\pm 1.11}$ | $73.10_{\pm 1.40}$ | $62.45_{\pm 0.69}$ |
| | $K = 5$ | $49.33_{\pm 1.03}$ | $65.67_{\pm 1.11}$ | $74.00_{\pm 1.39}$ | $63.00_{\pm 0.68}$ |
| 600M ensembles | $K = 1$ | $45.92_{\pm 1.02}$ | $62.84_{\pm 1.13}$ | $68.50_{\pm 1.47}$ | $59.09_{\pm 0.71}$ |
| | $K = 2$ | $47.56_{\pm 1.02}$ | $64.04_{\pm 1.12}$ | $71.80_{\pm 1.42}$ | $61.13_{\pm 0.69}$ |
| | $K = 3$ | $48.44_{\pm 1.03}$ | $64.25_{\pm 1.12}$ | $73.30_{\pm 1.40}$ | $62.00_{\pm 0.69}$ |
| | $K = 4$ | $49.03_{\pm 1.03}$ | $64.80_{\pm 1.11}$ | $73.70_{\pm 1.39}$ | $62.51_{\pm 0.69}$ |
| | $K = 5$ | $50.34_{\pm 1.03}$ | $64.80_{\pm 1.11}$ | $75.30_{\pm 1.36}$ | $63.48_{\pm 0.68}$ |
| 1.4B ensembles | $K = 1$ | $43.56_{\pm 1.02}$ | $64.20_{\pm 1.12}$ | $68.80_{\pm 1.47}$ | $58.85_{\pm 0.70}$ |
| | $K = 2$ | $47.26_{\pm 1.02}$ | $65.13_{\pm 1.11}$ | $75.30_{\pm 1.36}$ | $62.56_{\pm 0.68}$ |
| | $K = 3$ | $49.33_{\pm 1.03}$ | $65.40_{\pm 1.11}$ | $76.50_{\pm 1.34}$ | $63.74_{\pm 0.67}$ |
| | $K = 4$ | $48.86_{\pm 1.03}$ | $66.38_{\pm 1.10}$ | $77.80_{\pm 1.31}$ | $64.35_{\pm 0.67}$ |
| | $K = 5$ | $49.71_{\pm 1.03}$ | $66.38_{\pm 1.10}$ | $77.10_{\pm 1.33}$ | $64.39_{\pm 0.67}$ |
| Distillation (300M) | Self | $46.68_{\pm 1.02}$ | $62.35_{\pm 1.13}$ | $72.60_{\pm 1.41}$ | $60.54_{\pm 0.69}$ |
| | Ensemble | $48.44_{\pm 1.03}$ | $62.84_{\pm 1.13}$ | $75.30_{\pm 1.36}$ | $62.19_{\pm 0.68}$ |
| Model soups | $K = 2$ | $26.56_{\pm 0.91}$ | $54.84_{\pm 1.16}$ | $24.70_{\pm 1.36}$ | $35.37_{\pm 0.67}$ |
| | $K = 4$ | $24.96_{\pm 0.89}$ | $55.28_{\pm 1.16}$ | $23.90_{\pm 1.35}$ | $34.71_{\pm 0.66}$ |

Table 5: Benchmark accuracies of all methods using 200M tokens on ARC-Easy, PIQA, and SciQ with averages. Entries are value$_{\pm \mathrm{SE}}$ in percentage points.

## H    CONTINUED PRE-TRAINING

### H.1    HYPERPARAMETERS

Hyperparameters for continued pre-training baselines are shown in Table 6. The 73B CPT run uses the default hyperparameters from (Wang et al., 2025), except for learning rate which we tuned ourselves. The individual members of the $K$-ensembles use the same hyperparameters as the standard recipe.

### H.2    CPT SOUPS

We ablate the performance of model soups compared to ensembling in our continued pre-training setting by averaging the weights of the members instead of ensembling them. Unlike standard pre-training, CPT soups perform strongly and slightly outperform ensembles as we increase the number of averaged models (Table 7).

| Parameter | Default | Lower BS | Epoching |
|---|---|---|---|
| Learning Rate | 3e-5 | 3e-5 | 3e-5 |
| Weight Decay | 0.1 | 0.1 | 0.1 |
| Batch Size | 512 | 64 | 64 |
| Epochs | 1 | 1 | 4 |

Table 6: Hyperparameters for continued pre-training.

Table 7: Continually pre-trained ensembles vs. soups

| Benchmarks | Llama 3B | $K$-ensembles | | | $K$-soups | | |
|---|---|---|---|---|---|---|---|
| | | $K=2$ | $K=4$ | $K=8$ | $K=2$ | $K=4$ | $K=8$ |
| GSM8K$_{(8\text{-shot})}$ | 28.23 | 49.28 | 51.80 | 52.99 | 49.73 | 53.83 | **54.96** |
| MATH$_{(4\text{-shot})}$ | 6.90 | 21.84 | 23.04 | 23.50 | 22.40 | 23.02 | **23.72** |
| MATHQA$_{(8\text{-shot})}$ | 35.07 | 45.12 | 46.06 | 45.26 | 44.59 | **46.10** | 45.33 |
| Average | 24.25 | 38.79 | 40.35 | 40.58 | 38.91 | 40.98 | **41.34** |

# I  POWER LAWS

## I.1  SENSITIVITY ANALYSIS

To test whether our asymptote estimation is reliable due to run-to-run variance, we conduct a sensitivity analysis for regularized parameter scaling and ensembling, shown in Figure 20.

To test parameter scaling, we fit three power laws to all the models trained where each power law uses a different seed (governing data order and model initialization). Though the scaling laws change per seed, they remain relatively consistent, with the asymptotes staying close together. This is encouraging, as the standard deviation in asymptotes is close to the run-to-run standard deviation for 300M models (Appendix D.3).

Since we have more runs for ensembling, we test the reliability of ensembling by subsampling the number of members. When fitting a power law using up to four ensemble members, we find an extremely similar law to using up to eight ensemble members. Qualitatively, over the course of our experiments, we found that the scaling law for ensembling is a lot more stable than the scaling law for parameter scaling.

We note that this is a limited stress-test and that it is likely our asymptote estimation procedure is quite noisy. Furthermore, we note that this does not test our two-tier and three-tier power laws for joint scaling of parameters, members, and data, nor does it test our settings where our best run is further from the asymptote. We advise taking these asymptotes with a grain of salt and interpreting them as rough estimates.

## I.2  FITTING LAWS

To fit our power laws, we use `scipy.optimize.curve_fit`, either with no initial conditions and bounds or with `p0=[1.0, 0.5, 2.0]` and `bounds=([0, 0, 0], [np.inf, np.inf, np.inf])` '. We note that unlike prior work where such parameters have been found to be important (Besiroglu et al., 2024; Hoffmann et al., 2022), we did not find them to be critical considering how our fits are simple and over 1 dimension.

# J  ADDITIONAL RELATED WORK

**Over-parametrized machine learning.**   We show that even though early work in double descent suggests that over-parameterized deep learning does not have clean scaling laws due to double descent (Belkin et al., 2019; Hastie et al., 2020; Nakkiran et al., 2019), we can get clean scaling via

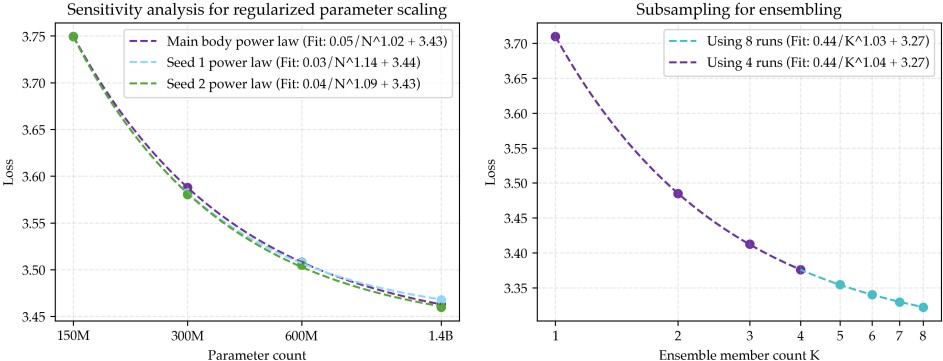

Figure 20: **Sensitivity analysis.** Left: When re-fitting the regularized power law across two additional seeds, we find that the asymptote stays relatively stable. Right: When subsampling the number of points for the ensemble scaling law, we find that the power law barely changes.

tuning regularization, agreeing with theory in over-parameterized regression (Advani and Ganguli, 2016; Nakkiran et al., 2021; Canatar et al., 2021; Simon et al., 2024).

**Distillation algorithms** Though we use sequence knowledge distillation (Kim and Rush, 2016) as a preliminary demonstration, there are many better distillation algorithms such as using more supervision via logits (Sanh et al., 2020) and minimizing different divergences between the teacher and student (Agarwal et al., 2024; Gu et al., 2024). Past work has further quantified the scaling properties of logit distillation (Busbridge et al., 2025), and logit distillation is increasingly used to pre-train the most performant small language models (Goyal et al., 2025; Yang et al., 2025; Team et al., 2025a).

**Synthetic data.** We can interpret distillation as a form of synthetic data, and unlike recent work on synthetic data to improve data efficiency (Maini et al., 2024; Allen-Zhu and Li, 2024; Su et al., 2025; Team et al., 2025b; Yang et al., 2024; Ruan et al., 2025), distillation requires minimal human priors such as a trusted reward function or known augmentation invariance like rephrasing.

**Classical data-constrained deep learning.** Historically, many machine learning benchmarks before the era of large language models were data-constrained (Marcus et al., 1993; Warstadt et al., 2023; Deng et al., 2009; Lecun et al., 1998), resulting in many effective algorithms. For example, the best models on Penn Tree Bank utilized dynamic evaluation (Mikolov et al., 2010; Krause et al., 2017), ensembling and model averaging (Takase et al., 2018; Zaremba et al., 2015), regularization (drop-out, weight decay, weight tying, lower batch size) (Merity et al., 2017; Zaremba et al., 2015; Gal and Ghahramani, 2016), data augmentation (Shi et al., 2021; Xie et al., 2017), and novel architectures (Zilly et al., 2017; Grave et al., 2016; Yang et al., 2018). We revisit a few of these algorithms and advocate for future work to explore all others.

## K UPDATES DURING REBUTTALS

### K.1 NEW MODEL SIZES

Our 1.4B parameter model is too shallow and does not follow a consistent aspect ratio compared to our smaller models. We pre-train 1.5B and 3.2B parameter models with fixed aspect ratios at the 200M token scale. We detail the new model configurations in Table 8. We show the losses of locally optimal regularized and unregularized models in Figure 21.

| Parameter | 150M | 300M | 600M | 1.4B | 1.5B | 3.2B |
|---|---|---|---|---|---|---|
| Context Length | 4096 | 4096 | 4096 | 4096 | 4096 | 4096 |
| Hidden Dimension | 512 | 768 | 1024 | 2048 | 1536 | 2048 |
| Intermediate Dimension | 1792 | 2688 | 3584 | 7168 | 5376 | 7168 |
| Attention Heads | 8 | 12 | 16 | 16 | 16 | 16 |
| KV Heads | 8 | 12 | 8 | 8 | 8 | 8 |
| Layers | 6 | 12 | 24 | 16 | 36 | 48 |

Table 8: New model architecture configurations for different model sizes.

### K.2 MIXTURE-OF-EXPERTS

We perform some intial experiments on the data efficiency of alternative architectures, specifically Mixture-of-Experts (Shazeer et al., 2017). We pre-train an MoE with 300M active parameters following standard recipes from Mixtral (Jiang et al., 2024) and OLMoE (Muennighoff et al., 2025). We use a token choice router, load balancing and z losses with default hyperparameters of 0.01 and 0.001 respectively, 2 active experts per token, and 8 total experts. See Table 9 for additional details.

We search for locally optimal hyperparameters of this architecture (including regularization) and achieve 3.57 loss, slightly outperforming the regularized 300M dense model with loss 3.58. This initial experiment suggests that the data efficiency of MoEs is similar to dense models, but we leave a more thorough study to future work.

| Parameter | Dense 300M | MoE Active 300M |
|---|---|---|
| Context Length | 4096 | 4096 |
| Hidden Dimension | 768 | 768 |
| Intermediate Dimension | 2688 | 1344 |
| Attention Heads | 12 | 12 |
| KV Heads | 12 | 12 |
| Layers | 12 | 12 |
| Experts per token | – | 2 |
| Total experts | – | 8 |

Table 9: Model architecture configuration for 300M dense vs. Mixture-of-Experts.

| Model | Val Loss | Learning Rate | Epoch Count | Weight Decay |
|---|---|---|---|---|
| Dense 300M | 3.5868 | 3e-3 | 16 | 1.6 |
| MoE Active 300M | 3.5767 | 3e-3 | 8 | 1.6 |

Table 10: Loss and hyperparameters of the 300M dense vs. Mixture-of-Experts.

| Scaling law | | $R^2$ |
|---|---|---|
| Figure 3: Regularized parameter scaling | 300M regularized recipe | 0.9999 |
| Figure 4: Ensemble scaling | 300M ensemble (standard hparams) | 0.9999 |
| Figure 5: Joint scaling | 150M ensemble (tuned) | 1.000 |
| | 300M ensemble (tuned) | 1.000 |
| | 600M ensemble (tuned) | 1.000 |
| | 1.4B ensemble (tuned) | 0.9998 |
| | Parameter scaling double limit | 0.9983 |
| Figures 6 and 7: Data scaling for all recipes | Standard recipe | 1.000 |
| | Regularized recipe | 0.9998 |
| | Joint scaling recipe | 0.9999 |

Table 11: $R^2$ for our main scaling laws.

### K.3 GOODNESS-OF-FIT AND EXTRAPOLATION

#### K.3.1 GOODNESS-OF-FIT

We report $R^2$ for the important scaling laws we display in our paper in Table 11. We note that all of these values are quite high (above 0.999), likely due to the low sample count. Nonetheless, these simple fits are much stronger than prior work that use even more expressive scaling laws (e.g. Muennighoff et al. (2023); Prabhudesai et al. (2025)).

#### K.3.2 EXTRAPOLATION

Due to the nature of our fits, we believe that the above $R^2$ analysis is insufficient to test our scaling laws. Therefore, we perform an extrapolation test to predict the losses a larger model. For this, we train 1.5B and 3.2B models following the config described in K.1 with locally optimal hyperparameters. We find that our regularized power law (purple dotted line) predicts the loss of the 1.5B model with error 0.005 and the 3.2B model with error 0.008, suggesting that our scaling laws have reasonable predictive power. Further, we find that larger unregularized models (red stars) eventually overfit, even when we fix the aspect ratio.

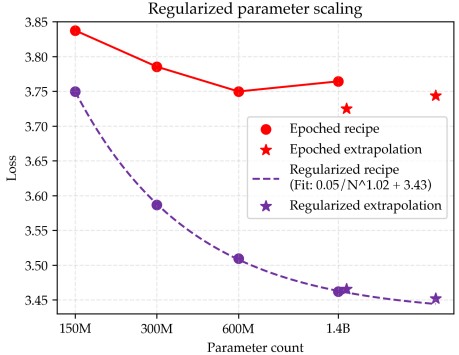

| Tuned $H$ | 150M | 300M | 600M | 1.4B | 1.5B | 3.2B |
|---|---|---|---|---|---|---|
| Learning rate | 3e-3 | 3e-3 | 1e-3 | 1e-3 | 1e-3 | 1e-3 |
| Epoch count | 16 | 16 | 8 | 8 | 8 | 8 |
| Weight decay | 0.8 | 1.6 | 3.2 | 3.2 | 3.2 | 3.2 |

Figure 21: **Extrapolation to larger model sizes.** We extrapolate the predictions of our power law from Figure 3. We find that our power law predicts the loss of the 1.5B model with error 0.005 and the 3.2B model with error 0.008. We also find that even when fixing aspect ratio, large enough models without regularization (red stars) overfit.

### K.4 MORE EVALUATIONS

We evaluate additional benchmarks to test the reliability of our improvements in validation loss. In addition to ARC-Easy, PIQA, and SciQ, we now evaluate ARC-Challenge, HellaSwag, LAMBADA, and Winogrande. We visualize the results in Figure 22 and find a consistent ranking over regularization, parameter scaling, ensemble scaling, and distillation. Additionally, the unregularized 1.5B and 3.2B models described in Appendix K.1 still underperform smaller models.

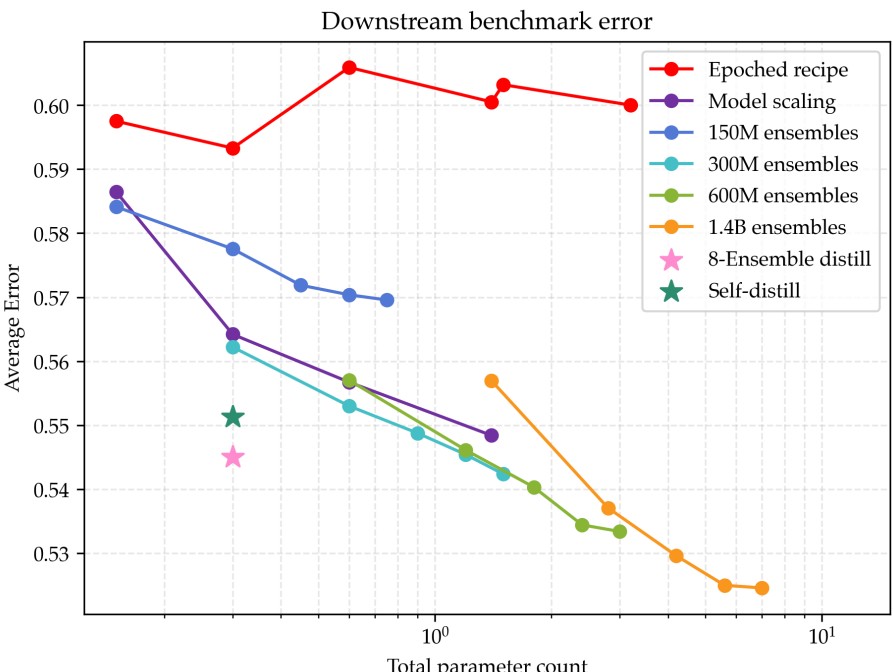

Figure 22: **Performance of pre-trained models on downstream tasks.** We re-evaluate our models and ensembles on 7 total downstream benchmarks. Models with lower validation loss have lower average error across downstream benchmarks.

| Model type | | ARC-Challenge (%) | ARC-Easy (%) | HellaSwag (%) | LAMBADA (%) | PIQA (%) | SciQ (%) | Winogrande (%) | Avg (%) |
|---|---|---|---|---|---|---|---|---|---|
| Unregularized model scaling | 150M | 18.69±1.14 | 40.95±1.01 | 26.62±0.44 | 23.93±0.59 | 59.52±1.15 | 62.00±1.54 | 50.04±1.41 | 40.25±0.42 |
| | 300M | 19.45±1.16 | 41.75±1.01 | 27.75±0.45 | 23.09±0.59 | 61.04±1.14 | 62.50±1.53 | 49.09±1.41 | 40.67±0.42 |
| | 600M | 17.32±1.11 | 40.11±1.01 | 27.00±0.44 | 21.09±0.57 | 59.90±1.14 | 60.40±1.55 | 50.04±1.41 | 39.41±0.41 |
| | 1.4B | 18.00±1.12 | 40.49±1.01 | 27.40±0.45 | 21.93±0.58 | 60.07±1.14 | 61.00±1.54 | 50.75±1.41 | 39.95±0.42 |
| | 1.5B | 18.09±1.12 | 41.96±1.01 | 27.39±0.45 | 21.04±0.57 | 61.21±1.14 | 57.00±1.57 | 51.07±1.40 | 39.68±0.42 |
| | 3.2B | 18.94±1.15 | 40.95±1.01 | 27.76±0.45 | 21.50±0.57 | 61.21±1.14 | 59.50±1.55 | 50.12±1.41 | 40.00±0.42 |
| Model scaling | 150M | 19.37±1.15 | 41.20±1.01 | 27.01±0.44 | 26.55±0.62 | 60.28±1.14 | 64.00±1.52 | 51.07±1.40 | 41.35±0.42 |
| | 300M | 19.80±1.16 | 44.40±1.02 | 28.65±0.45 | 28.95±0.63 | 62.19±1.13 | 69.30±1.46 | 51.78±1.40 | 43.58±0.41 |
| | 600M | 19.88±1.17 | 47.14±1.02 | 29.16±0.45 | 31.81±0.65 | 62.68±1.13 | 69.60±1.46 | 50.04±1.41 | 44.33±0.41 |
| | 1.4B | 20.14±1.17 | 45.83±1.02 | 30.17±0.46 | 32.66±0.65 | 63.87±1.12 | 72.70±1.41 | 50.75±1.41 | 45.16±0.41 |
| 150M ensembles | $K=1$ | 18.94±1.15 | 42.85±1.02 | 27.34±0.44 | 25.62±0.61 | 60.88±1.14 | 64.90±1.51 | 50.59±1.41 | 41.59±0.42 |
| | $K=2$ | 19.54±1.16 | 44.53±1.02 | 27.13±0.44 | 25.11±0.60 | 62.13±1.13 | 65.90±1.50 | 51.38±1.40 | 42.25±0.42 |
| | $K=3$ | 19.54±1.16 | 44.95±1.02 | 27.27±0.44 | 25.42±0.61 | 61.97±1.13 | 68.30±1.47 | 52.25±1.40 | 42.81±0.41 |
| | $K=4$ | 18.94±1.15 | 45.83±1.02 | 27.42±0.45 | 25.52±0.61 | 62.02±1.13 | 69.30±1.46 | 51.70±1.40 | 42.96±0.41 |
| | $K=5$ | 19.28±1.15 | 45.41±1.02 | 27.39±0.45 | 25.58±0.61 | 62.57±1.13 | 70.40±1.44 | 50.67±1.41 | 43.04±0.41 |
| 300M ensembles | $K=1$ | 21.42±1.20 | 44.28±1.02 | 29.54±0.46 | 28.08±0.63 | 63.06±1.13 | 68.00±1.48 | 52.09±1.40 | 43.78±0.42 |
| | $K=2$ | 20.99±1.19 | 46.55±1.02 | 29.37±0.45 | 30.06±0.64 | 63.98±1.12 | 70.50±1.44 | 51.46±1.40 | 44.70±0.41 |
| | $K=3$ | 20.82±1.19 | 47.81±1.02 | 29.45±0.45 | 30.12±0.64 | 64.69±1.12 | 72.70±1.41 | 50.28±1.41 | 45.12±0.41 |
| | $K=4$ | 19.80±1.16 | 48.23±1.03 | 29.30±0.45 | 30.25±0.64 | 65.56±1.11 | 73.30±1.40 | 51.78±1.40 | 45.46±0.41 |
| | $K=5$ | 19.45±1.16 | 49.41±1.03 | 29.51±0.46 | 30.39±0.64 | 65.89±1.11 | 74.00±1.39 | 51.70±1.40 | 45.76±0.41 |
| 600M ensembles | $K=1$ | 21.08±1.19 | 46.00±1.02 | 31.03±0.46 | 28.76±0.63 | 62.30±1.13 | 68.60±1.47 | 52.33±1.40 | 44.30±0.42 |
| | $K=2$ | 21.67±1.20 | 47.60±1.02 | 31.20±0.46 | 30.89±0.64 | 64.15±1.12 | 71.70±1.43 | 50.51±1.41 | 45.39±0.41 |
| | $K=3$ | 21.67±1.20 | 48.27±1.03 | 31.00±0.46 | 32.00±0.65 | 64.25±1.12 | 73.20±1.40 | 51.38±1.40 | 45.97±0.41 |
| | $K=4$ | 22.61±1.22 | 48.95±1.03 | 31.03±0.46 | 32.62±0.65 | 64.42±1.12 | 73.80±1.39 | 52.49±1.40 | 46.56±0.41 |
| | $K=5$ | 20.99±1.19 | 50.51±1.03 | 31.12±0.46 | 32.80±0.65 | 64.74±1.11 | 75.10±1.37 | 51.38±1.40 | 46.66±0.41 |
| 1.4B ensembles | $K=1$ | 21.67±1.20 | 43.60±1.02 | 31.88±0.47 | 27.38±0.62 | 64.25±1.12 | 68.80±1.47 | 52.57±1.40 | 44.31±0.42 |
| | $K=2$ | 21.67±1.20 | 47.35±1.02 | 31.90±0.47 | 31.54±0.65 | 65.13±1.11 | 75.10±1.37 | 51.38±1.40 | 46.29±0.41 |
| | $K=3$ | 21.42±1.20 | 49.24±1.03 | 32.15±0.47 | 32.76±0.65 | 65.61±1.11 | 76.40±1.34 | 51.70±1.40 | 47.04±0.41 |
| | $K=4$ | 22.35±1.22 | 49.12±1.03 | 32.41±0.47 | 33.65±0.66 | 65.89±1.11 | 77.80±1.31 | 51.30±1.40 | 47.50±0.41 |
| | $K=5$ | 21.08±1.19 | 49.79±1.03 | 32.51±0.47 | 34.00±0.66 | 66.32±1.10 | 77.10±1.33 | 52.01±1.40 | 47.54±0.41 |
| Distillation (300M) | Self | 20.05±1.17 | 46.68±1.02 | 28.97±0.45 | 32.04±0.65 | 62.46±1.13 | 72.30±1.42 | 51.62±1.40 | 44.87±0.41 |
| | Ensemble | 19.20±1.15 | 48.15±1.03 | 28.54±0.45 | 32.52±0.65 | 62.51±1.13 | 75.10±1.37 | 52.49±1.40 | 45.50±0.41 |

Table 12: Benchmark accuracies of all methods using 200M tokens on ARC-Challenge, ARC-Easy, HellaSwag, LAMBADA, PIQA, SciQ, and Winogrande. Entries are value$_{\pm\text{SE}}$ in percentage points.