# OpenReview forum: "Pre-training under infinite compute"
_ICLR.cc/2026/Conference — ICLR 2026 Oral_

### Official Review · Reviewer_7Xg1 · 2025-10-20

**Soundness:** 3
**Presentation:** 3
**Contribution:** 3
**Rating:** 6
**Confidence:** 3

**Summary:**

The paper explores a forward-looking problem: how to design pre-training strategies when compute grows faster than data availability. The authors simulate a future where web text becomes the limiting resource and propose several methods to make pre-training data-efficient under fixed data but infinite compute.

**Strengths:**

1. The paper identifies an under-explored but realistic future scenario where compute scales faster than data. This is a fresh take beyond the traditional “compute-optimal” (Chinchilla-style) scaling laws.
2. The authors construct a controlled 200M-token benchmark and systematically vary parameters, epochs, and regularization. The coordinate-descent tuning of weight decay, LR, and epochs shows strong empirical rigor.
3. The evaluation results demonstrate the effectiveness of the proposed approach.

**Weaknesses:**

1. The question raised in this paper is quite interesting. However, I am curious whether the conclusion would still hold if the dataset size were fixed between 30B and 100B tokens while training a 1B-parameter model. According to the Chinchilla paper, the optimal number of training tokens is approximately 20 x N, and over-training is quite common nowadays. For instance, the Qwen3 model was pretrained on about 36T tokens. It would be interesting to investigate, when training with more than 20 x N tokens, how to appropriately adjust the learning rate, weight decay, and number of epochs.
2. Could the authors briefly clarify how this paper differs from the Tensor Programs V: Tuning Large Neural Networks via Zero‑Shot Hyperparameter Transfer (https://arxiv.org/pdf/2203.03466)? Additionally, the μP paper shows that hyperparameters are closely tied to the model architecture, and the more recent Scaling Inference‑Efficient Language Models (https://arxiv.org/pdf/2501.18107) builds a scaling law that explicitly incorporates architecture. Could the authors elaborate on the impact of model architecture in their study?
3. The evaluation over three downstream tasks is limited. Is it possible for the authors to show the evaluation over Arc-Challenge, Lambada, HellaSwag, Winogrande, etc?

I would be willing to raise my score if the authors can address the above questions.

**Questions:**

See weaknesses

---

> ### Author Response · Authors · 2025-11-21
>
> Thank you for your positive review of our work! We respond to the remaining concerns below.
>
> ---
> > The question raised in this paper is quite interesting. However, I am curious whether the conclusion would still hold if the dataset size were fixed between 30B and 100B tokens while training a 1B-parameter model. According to the Chinchilla paper, the optimal number of training tokens is approximately 20 x N, and over-training is quite common nowadays. For instance, the Qwen3 model was pretrained on about 36T tokens. It would be interesting to investigate, when training with more than 20 x N tokens, how to appropriately adjust the learning rate, weight decay, and number of epochs.
>
> This is an interesting question addressed by our paper! In Appendix C.3 and Figure 11, we provide a comprehensive analysis about how the hyperparameters change as a function of the data-to-parameter ratio. As a brief summary:
> - Optimal learning rate mostly depends on architecture, not data-to-compute
> - Optimal epoch count increases for higher data-to-parameter ratio (i.e. smaller models or more data)
> - Optimal weight decay decreases for higher data-to-parameter ratio (i.e. smaller models or more data)
>
> Our regime with the highest data-to-parameter ratio is close to Chinchilla (150M models for 1.6B tokens), and we find that the optimal epoch count is relatively high (64) and the optimal weight decay is current practice of 0.1. Therefore, we expect that for the parameter-constrained regime you described, the main hyperparameter difference from standard practice is epoch count and not learning rate or weight decay. On the other hand, weight decay is critical for the realistic data-constrained future.
> More broadly, we find that to train the best possible model of smaller parameter count, it is better to first train a large model/ensemble and distill it into a small model as opposed to training the small model directly (discussed in Section 6). This is in line with current practice of how the most performant small models are trained (e.g. Qwen3 family).
>
> > Could the authors briefly clarify how this paper differs from the Tensor Programs V: Tuning Large Neural Networks via Zero‑Shot Hyperparameter Transfer (https://arxiv.org/pdf/2203.03466)? Additionally, the μP paper shows that hyperparameters are closely tied to the model architecture, and the more recent Scaling Inference‑Efficient Language Models (https://arxiv.org/pdf/2501.18107) builds a scaling law that explicitly incorporates architecture. Could the authors elaborate on the impact of model architecture in their study?
>
> Tensor Programs V foremost gives a scaling recipe for setting learning rate and initializer scale as we increase model width. We can not use this heuristic directly since (1) it does not apply to depth scaling (2) it does not describe the interaction between learning rate + weight decay and overfitting in the epoching regime (3) it underperforms standard variance scaling in our pre-training framework (https://github.com/marin-community/marin/issues/621).
> Thanks for sharing the interesting work surrounding architecture! In relation to this work, our approach tackles training inference efficient models by distilling large models/ensembles as detailed in Section 6. More broadly, different architectures have different data efficiencies and optimal hyperparameters, and our framework offers a clear prescription for how to compare such architecture against each other. We also conduct new experiments suggesting that MoE’s share a similar data-efficiency to their dense counterparts, with a full elaboration given in Appendix K.2.We are happy to include a longer discussion on architecture in the main body to clarify its interaction with data efficiency.
>
> > The evaluation over three downstream tasks is limited. Is it possible for the authors to show the evaluation over Arc-Challenge, Lambada, HellaSwag, Winogrande, etc?
>
> We re-run our evaluations on all of our models including the mentioned benchmarks for a total of 7 downstream tasks (ARC-Challenge, ARC-Easy, HellaSwag, Lambada, PIQA, SciQ, Winogrande). We find that all of the trends surrounding regularization, parameter scaling, ensemble scaling, and distillation are preserved, suggesting that our evaluation based results are robust. In Appendix K.5, we share the evaluations both as a plot and full table decomposition.
>
> ---
>
> Thank you again for your review, does the above address your concerns? Please let us know if you have any further questions!

---

### Official Review · Reviewer_rfVe · 2025-10-30

**Soundness:** 4
**Presentation:** 4
**Contribution:** 4
**Rating:** 8
**Confidence:** 4

**Summary:**

This paper investigates language model pre-training in a fixed data infinite compute setting. To address overfitting in the data-constrained scenario, the authors propose a recipe with 30x larger weight decay and the asymptote of the scaling law. The final joint scaling recipe yields a 5x data efficiency gain over the baseline.

The authors then propose and demonstrate that model ensembles can be trained and distilled into a single student model while preserving most of the loss improvement.

Lastly, the authors show that their metrics generalize well to downstream benchmarks.

**Strengths:**

1. The motivation for the paper is practical, where internet data is exhausted.
2. The empirical studies are thorough and clear. The authors produced a recipe that achieves 5x higher data efficiency and demonstrates the benefit of a much larger weight decay.
3. The level of attention to experiment details is commendable, and experiment setups are described in detail.
4. The settings the authors take into account are comprehensive, with results ranging from pretraining, model ensembling, and distillation.

**Weaknesses:**

1. Due to the shallow setup for the 1.4B model, it underperforms the 600M version without the proposed weight decay. While in Figure 3, the authors show how the regularized recipe fixes this, this still casts doubt on the following conclusions, given how sensitive power law fits are, and that 1.4B is the largest model variant that the author uses for curve fitting.  For instance, in Figure 9, 1.4B's downstream eval results also break the trend, most likely due to model architecture configuration. Since the authors only train on 200M tokens, this is not prohibitively expensive to fix.

2. The ensemble scaling part requires a heuristic that breaks in the over-parametrized setting. The heuristic itself, when and why it works well or not, could use more explanation to make the ensemble tuning recipe simpler and more principled.

**Questions:**

1. Is there any intuition behind distilling from a single model working better than distilling the ensemble?

---

> ### Author Response · Authors · 2025-11-21
>
> Thank you for your positive review of our work! We respond to the remaining concerns below.
>
> ---
>
> > Due to the shallow setup for the 1.4B model, it underperforms the 600M version without the proposed weight decay. While in Figure 3, the authors show how the regularized recipe fixes this, this still casts doubt on the following conclusions, given how sensitive power law fits are, and that 1.4B is the largest model variant that the author uses for curve fitting. For instance, in Figure 9, 1.4B's downstream eval results also break the trend, most likely due to model architecture configuration. Since the authors only train on 200M tokens, this is not prohibitively expensive to fix.
>
> 1. Thank you for the astute observation about the 1.4B model! We rectify this error by finding locally optimal hyperparameters for a 1.5B model where we set the width/depth in a fixed aspect ratio following the 150M, 300M, and 600M (new configs in Appendix K.1). We find that our power law is still predictive of this point, predicting the correct loss with an error of 0.005.
> 2. We further stress test the extrapolation of our scaling laws by training a locally optimal 3.2B model with the same width/depth aspect ratio as the smaller models. We find that we also accurately predict this model’s loss with an error of only 0.008. This is quite low considering the run-to-run standard deviation of 0.008.
> 3. We re-run downstream evals on a larger set of 7 pre-training benchmarks with the newly trained 1.5B and 3.2B models in Appendix K.5. We find that even with proper width/depth scaling, unregularized epoched models with locally optimal hyperparameters (red line) perform poorly on downstream benchmarks.
>
> > The ensemble scaling part requires a heuristic that breaks in the over-parametrized setting. The heuristic itself, when and why it works well or not, could use more explanation to make the ensemble tuning recipe simpler and more principled.
>
> Thank you for bringing this to our attention! We provide a detailed discussion of the heuristic and its success in Appendix D.4 due to the page limit. Briefly, we find that across all of our settings, it is better to use double the epochs and half the weight decay relative to the optimal regularized hyperparameters. For our 200M setting, we are able to certify this is locally optimal (with only one exception). Was there anything we could add to this section to make it more comprehensive? We can bring it into the main body after the page limit increases.
>
> > Is there any intuition behind distilling from a single model working better than distilling the ensemble?
>
> We apologize for any confusion in the writing: distilling from an ensemble (pink star in Figure 8) outperforms self-distillation (teal star in Figure 8). However, self-distillation outperforms training arbitrarily larger models, which we predict is due to the multi-view nature of data as described in [Allen-Zhu and Li, 2023](https://arxiv.org/abs/2012.09816). Happy to make this more clear!
>
> ---
>
> Thank you again for your review, does the above address your concerns? Please let us know if you have any further questions!

---

> > ### Comment · Reviewer_rfVe · 2025-11-25
> >
> > Thank you for the detailed response. I will maintain my positive score.

---

### Official Review · Reviewer_GYnT · 2025-10-30

**Soundness:** 3
**Presentation:** 4
**Contribution:** 4
**Rating:** 8
**Confidence:** 3

**Summary:**

This paper presents a compelling and timely investigation into language model pre-training for a future where high-quality data is the primary bottleneck, not compute. The authors argue that as compute continues to scale faster than data generation, the community must shift from compute-efficient to data-efficient training paradigms. The work first demonstrates that standard recipes, which involve increasing epochs or model parameters, inevitably lead to overfitting and performance degradation. As a solution, the authors propose a "regularized recipe" using aggressive weight decay (up to 30x standard practice) to achieve stable, monotonic performance scaling. Building on this, they show that ensembling multiple smaller models is a more data-efficient use of compute than scaling a single large model. By combining these methods, they claim a significant 5.17x improvement in data efficiency. Finally, the paper grounds these compute-intensive findings in practice, showing that the gains from ensembling can be largely preserved by distilling them into a single, smaller model, and that improvements in validation loss translate to meaningful gains on downstream tasks.

**Strengths:**

This is a strong paper that could become a foundational empirical work for the emerging data-constrained training regime. The experiments are thorough, and the insights are valuable for both theoretical and practical applications. Specifically, I appreciate the following contributions of this paper.
1. **Extensive Experiments:** The claims are backed by an impressive scale of experimentation, reportedly over 2000 training runs. I like the scientific approach of diagnosing the failure of a baseline, introducing a robust regularized recipe, and then demonstrating the superiority of ensembling, which builds a convincing, step-by-step argument.
2. **Practical and Generalizable Insights:** The work successfully connects the "infinite compute" thought experiment back to practical applications. The demonstration that an 8-member ensemble can be distilled into a single student model that retains 83% of the performance gain is a crucial result, proving these methods are not purely theoretical. Furthermore, the validation on downstream benchmarks  confirms that the observed improvements in pre-training loss correspond to genuine enhancements in model capabilities.

**Weaknesses:**

1. **Architectural Specificity:** The empirical results are derived exclusively from Llama-style decoder-only architectures. It remains unclear whether the core findings, particularly the superiority of ensembling over scaling a single model, generalize to different architectures such as Mixture-of-Experts (MoE). More critically, under the infinite-compute regime, transformer architectures may not represent the optimal choice. The paper would benefit from discussion on architectural considerations within data-constrained training regimes.
2. **Reliability of Power-Law Extrapolation:** The paper's headline quantitative claims (e.g., 5.17× data efficiency) depend on extrapolating power laws fitted to limited data points (typically four model sizes). While the authors provide a sensitivity analysis and appropriately advise caution, the absence of formal statistical goodness-of-fit metrics (such as R² or confidence intervals) makes it difficult to assess the true reliability of these asymptotic estimates. The broader literature demonstrates that simple power laws can break down or exhibit sub-scaling behavior at larger scales, making long-range extrapolation risky. This concern is compounded by the non-standard width/depth configuration of the 1.4B model, which introduces a potential confounder that could affect the quality of the fit.

**Questions:**

The paper convincingly shows that tuned regularization eliminates the initial overfitting peak seen in standard recipes. However, the double descent literature suggests that with sufficient over-parameterization, an unregularized model's performance might enter a "second descent" and improve again. Could you comment on whether this second descent could, in the infinite limit, outperform the regularized recipe? Is there a theoretical or empirical reason to believe the regularized performance curve is strictly superior at all points in the over-parameterized regime?

---

> ### Author Response · Authors · 2025-11-21
>
> Thank you for your positive review of our work! We respond to the remaining concerns below.
>
> ---
>
> > Architectural Specificity: The empirical results are derived exclusively from Llama-style decoder-only architectures. It remains unclear whether the core findings, particularly the superiority of ensembling over scaling a single model, generalize to different architectures such as Mixture-of-Experts (MoE). More critically, under the infinite-compute regime, transformer architectures may not represent the optimal choice. The paper would benefit from discussion on architectural considerations within data-constrained training regimes.
>
> It is a good insight that the optimal architecture will likely change under the infinite compute regime. Though we focus more on ensembling/regularization/distillation, we perform some initial explorations with alternative architectures.
>
> 1. We include new experiments to test whether MoE’s improve data-efficiency. To test this, we train an MoE with 300M active parameters (2 active / 8 total experts) following standard recipes from Mixtral and OLMoE (e.g. token choice router, load balancing and z loss), see Appendix K.2 for more details. We search for locally optimal hyperparameters of this architecture (including regularization) and achieve 3.57 loss, slightly outperforming the regularized 300M dense model with loss 3.58. This initial experiment suggests that the data efficiency of MoEs is similar to dense models, but we leave a more thorough study to future work.
> 2. In Appendix D.5.1, we share how training a model with an ensemble architecture from scratch barely outperforms training a single model and significantly underperforms training a model with 10x parameters. This experiment, in line with the theory and experiments in [Allen-Zhu and Li, 2023](https://arxiv.org/abs/2012.09816), suggests that the principle of sparsity might not improve data efficiency.
>
> Beyond our initial experiments, we believe there is a lot of opportunity for better architecture in the data-efficient world, which can be evaluated by our asymptote framework. We are happy to include a longer discussion on architecture to clarify its interaction with data efficiency.
>
> > Reliability of Power-Law Extrapolation: The paper's headline quantitative claims (e.g., 5.17× data efficiency) depend on extrapolating power laws fitted to limited data points (typically four model sizes). While the authors provide a sensitivity analysis and appropriately advise caution, the absence of formal statistical goodness-of-fit metrics (such as R² or confidence intervals) makes it difficult to assess the true reliability of these asymptotic estimates. The broader literature demonstrates that simple power laws can break down or exhibit sub-scaling behavior at larger scales, making long-range extrapolation risky. This concern is compounded by the non-standard width/depth configuration of the 1.4B model, which introduces a potential confounder that could affect the quality of the fit.
>
> Thanks for your suggestions on how to better quantify our fits!
>
> 1. Thank you for the astute observation about the 1.4B model! We rectify this error by finding locally optimal hyperparameters for a 1.5B model where we set the width/depth in a fixed aspect ratio following the 150M, 300M, and 600M (new configs in Appendix K.1). We find that our power law is still predictive of this point, predicting the correct loss with an error 0.005.
> 2. We now test our scaling laws in two different manners. First, we measure goodness-of-fit via $R^2$ and report this for all of our scaling laws in Appendix K.3.1, finding that our $R^2$ is always above 0.99. Since $R^2$ is not a stringent test given few samples, we test extrapolation by training a locally optimal 3.2B model with the same width/depth aspect ratio as the smaller models. We find that we also perform a strong prediction of this model’s loss with error 0.008. This is quite close considering the standard deviation of run-to-run variance itself is itself 0.008.
> 3. Though, our main headline claim of 5.17x data efficiency does rely on extrapolation, we also report data efficiency improvements that require *no* extrapolation. Specifically, when training a 5 ensemble of 1.4B models, we find that its loss **interpolates** to a 3.75x data efficiency improvement over the standard recipe. Since there is no extrapolation involved in this calculation, it's convincing that we have actually achieved a large data efficiency win, and could achieve a larger win if we were willing to spend more compute.

---

> ### Author Response · Authors · 2025-11-21
>
> > The paper convincingly shows that tuned regularization eliminates the initial overfitting peak seen in standard recipes. However, the double descent literature suggests that with sufficient over-parameterization, an unregularized model's performance might enter a "second descent" and improve again. Could you comment on whether this second descent could, in the infinite limit, outperform the regularized recipe? Is there a theoretical or empirical reason to believe the regularized performance curve is strictly superior at all points in the over-parameterized regime?
>
> This is an interesting theoretical question! To start, since we tune for the optimal weight decay at every parameter count, by definition, our regularized recipe can only outperform the standard unregularized recipe. This leaves the additional question of whether we expect the standard recipe to ever match the performance of the regularized recipe. Theoretical results in this regime are mixed: for the latent space model in [Hastie et al, 2020](https://arxiv.org/abs/1903.08560), the unregularized recipe eventually matches the optimal regularized loss. However, in the ridge model and misspecified models, the unregularized recipe plateaus at a much worse performance than the regularized recipe. In conjunction with our empirical results, we do not have a strong reason to believe that the standard recipe will eventually meet the performance of the regularized recipe.
>
> ---
>
> Thank you again for your review, does the above address your concerns? Please let us know if you have any further questions!

---

> ### Comment · Reviewer_GYnT · 2025-11-28
>
> Thank you for the detailed response. I really appreciate all the new experiments with MoE, training 1.5B and 3.2B models, and adding goodness-of-fit. I don't have any remaining concerns, and I would be happy to see a longer discussion on architecture to clarify its interaction with data efficiency included in the revised manuscript. I will keep my positive score.

---

### Official Review · Reviewer_kN1H · 2025-11-01

**Soundness:** 3
**Presentation:** 3
**Contribution:** 3
**Rating:** 8
**Confidence:** 3

**Summary:**

This work studies how to pre-train when compute is abundant but data is limited. The authors show that standard approaches of increasing epochs or model size overfit, and that stronger regularization, especially much larger weight decay, helps sustain gains. They further show that ensembling independently trained models improves the loss asymptote, and that their best setup, combining regularization, epoching, and ensembling achieves the same performance with 5× less data. Finally, they find that most of their ensembling recipe benefit can be retained through distillation into a smaller model, with gains also reflected on downstream tasks.

**Strengths:**

The problem addressed is timely. The main ideas like stronger regularization, evaluating by scaling-law asymptotes, and using simple ensembling, are easy to follow and reproduce. The experiments consistently show clear gains and improved data efficiency, and the distillation results make the approach more useful. The paper is well-written, with clear experiments and a natural, convincing link between the sections.

**Weaknesses:**

See Questions section below.

**Questions:**

1. The novelty feels incremental in places. Ensembling and stronger regularization are well‑known tools; the main new piece seems to be “optimize for and compare asymptotes under fixed data.” Could you sharpen the claim of conceptual novelty beyond careful tuning + ensembling, and clarify what, if anything, is surprising in the results relative to prior data‑constrained scaling work?
2. Compute‑matching across alternatives could be clearer. For a fixed training FLOP budget, how do (i) a single larger model, (ii) K‑member ensembling, and (iii) spending compute to synthesize more training text compare?
3. Weight decay is pushed high and clearly helps, but other classic knobs (dropout, data noising, etc.) are not explored. Would be interested in knowing your thoughts on whether similar monotone‑scaling and asymptote reductions hold with these (or other) alternatives?

---

> ### Author Response · Authors · 2025-11-21
>
> Thank you for your positive review of our work! We respond to the remaining concerns below.
>
> ---
>
> > The novelty feels incremental in places. Ensembling and stronger regularization are well‑known tools; the main new piece seems to be “optimize for and compare asymptotes under fixed data.” Could you sharpen the claim of conceptual novelty beyond careful tuning + ensembling, and clarify what, if anything, is surprising in the results relative to prior data‑constrained scaling work?
>
> We appreciate this opportunity to clarify the sources of methodological novelty within our work.
>
> - Self-distillation: We believe that our application of self-distillation defies a lot of current intuition surrounding generative modeling and synthetic data. For example, recent LM work on training on self-generations (e.g. [Dohmatob et al, 2024](https://arxiv.org/abs/2410.04840), [Gerstgrasser et al, 2024](https://arxiv.org/abs/2404.01413)) characterizes how much it *hurts* and do not offer a demonstration that it is more data-efficient, especially when unconstrained by compute. Though self-distillation has been considered in the past (as discussed in the related work), our demonstration of its success is novel for the pre-training setting.
> - Double/triple limit scaling laws: Though scaling laws are known to take the limit as one resource approaches infinity (compute, data, etc.), we introduce tiered scaling laws for taking parameter count, ensemble member count, and token count jointly to infinity. We elaborate upon the careful design and relation to prior work in Appendix D.6.
> - Over-parameterized models: Departing from standard scaling laws, our models are more data-efficient thanks to being significantly over-parameterized relative to prior work, utilizing parameter-to-data ratios that are 140x Chinchilla-optimal and 14000x standard inference-constrained models.
> - Clarifying careful tuning: Though careful hyperparameter tuning may seem like the bare minimum, our weight decay tuning is quite different from standard practice: we are not aware of any language model pre-training run that uses weight decay over 0.1 with AdamW, much less our optimal selection of 3.2. Our final recipes offer significant improvements and qualitatively different conclusions relative to prior data-constrained recipes such as [Muennighoff et al, 2023](https://arxiv.org/abs/2305.16264).
> - Clarifying ensembling: Though ensembles are well-known to improve performance, there are mixed opinions about whether it outperforms parameter scaling: while [Lobacheva et al, 2020](https://arxiv.org/abs/2007.08483) shows ensembling is better, [Vyas et al, 2023](https://arxiv.org/abs/2305.18411) and [Ruben et al, 2024](https://arxiv.org/abs/2412.05418) claim there is no benefit. Our work, the first rigorous exploration of ensembles for language model pre-training, offers a clear answer to this important question.
>
> > Compute‑matching across alternatives could be clearer. For a fixed training FLOP budget, how do (i) a single larger model, (ii) K‑member ensembling, and (iii) spending compute to synthesize more training text compare?
>
> There is an interesting tradeoff when comparing the models at different compute thresholds. We provide a full tradeoff with respect to parameter count (i.e. inference FLOPs) in Figure 9, left (note that for models trained from scratch, the inference FLOPs roughly corresponds to train FLOPs). We find that for sufficiently low inference FLOPs, it is best to scale parameter count. Eventually, it becomes better to train 300M ensembles, followed by 600M ensembles, and finally 1.4B ensembles. Furthermore, distillation achieves remarkably high performance with low inference FLOPs. This offers clear recommendations for which method is best depending on a fixed parameter budget.
>
> > Weight decay is pushed high and clearly helps, but other classic knobs (dropout, data noising, etc.) are not explored. Would be interested in knowing your thoughts on whether similar monotone‑scaling and asymptote reductions hold with these (or other) alternatives?
>
> Broadly, we are optimistic about disjoint techniques for data efficiency composing with each other, which we demonstrate with weight decay and ensembling. We expect other knobs like dropout to boost performance on top of our existing interventions and believe it offers interesting future work. We also expect these to fit within the monotone scaling laws and asymptotes so long as the values are appropriately tuned, in line with theoretical and empirical expectations surrounding proper regularization.
>
> ---
>
> Thank you again for your review, does the above address your concerns? Please let us know if you have any further questions!

---

### Author Response · Authors · 2025-11-21

We thank the reviewers for their helpful feedback and appreciation of our work! In the individual responses, we address any remaining concerns; we have added an Appendix K to our paper that contains all of our new experiments, analyses, and revisions. As a brief summary:

- We train an MoE model with locally optimal hyperparameters to test alternative architectures and find that it achieves a loss similar to a dense counterpart with the same number of active parameters.
- We retrain 1.5B models with locally optimal hyperparameters that reflect fixed width/depth scaling consistent with our smaller models, finding similar scaling to the 1.4B model.
- We pretrain 3.2B models with locally optimal hyperparameters that reflect fixed width/depth scaling consistent with our scaling models.
- We quantify our goodness-of-fit with $R^2$ (which is consistently very high) and an extrapolation test which predicts the loss of the best 3.2B model with low error.
- We evaluate our models on 7 downstream evaluations, finding qualitatively similar results to our previous set of 3 evaluations.

Please let us know if there are any further questions, and thanks again for the helpful reviews!

---

### Comment · Area_Chair_JLwy · 2025-11-28

Dear Reviewers,

The discussion phase is now underway, and the authors have finished uploading their responses to reviewers. If you haven't already, please carefully review the authors' responses to understand their perspectives. Engage in thoughtful, constructive discussions with authors, sharing your thoughts and seeking clarifications. Please also update your review or rating if necessary.

It is noted in the guideline that reviewers can leave comments visible to authors **until Dec 2 11:59pm AoE**. Your active participation and contribution to the ongoing discussion are highly encouraged. Thank you very much for your contribution to ICLR.

Best regards,

AC

---

### Meta-Review · Area_Chair_RHXW · 2026-01-05

**Summary:**

The authors investigate how to maximize model performance when the amount of data is fixed but compute is unconstrained. They present some important key insights from their analysis. Reviewers primarily questioned whether the findings are actually general, asking whether the reported efficiency gains held for different architectures (e.g., MoEs), and a broader set of downstream benchmarks.

Some concerns about the methodology used were also raised: around reliability of extrapolating power laws from limited data points, specifically noting that the 1.4B model's non-standard configuration might skew the results. The authors addressed these issues by training new models with corrected aspect ratios, adding MoE experiments, and expanding evaluations, which reinforced the validity of their results.

**Reviewer Concerns:**

Addressed:

Reviewers GYnT and rfVe questioned the validity of extrapolating power laws from only four data points, particularly given the potential confounder of the 1.4B model's non-standard architecture. This was addressed by adding new experiemnts.

Some reviewers were concerned that the 1.4B model was too shallow. The authors ran an additional experiment on a similar size model controlling for the aspect ratio.

Reviewer's GYnT concerns were addressed with new MoE results.

Reviewer's 7Xg1 concerns were addressed by adding additional benchmarks.

Not addressed: Reviewer kN1H asked if similar benefits would hold with other training "tricks" like dropout or data noising, which were not explored in the submission.

**Reviewer Scores:**

I would expect all reviewers to maintain or increase their already-strong scores, with Reviewer 7Xg1 likely increasing.

---

### Decision · Program_Chairs · 2026-01-26

Accept (Oral)